# Cryo-EM reveals evolutionarily conserved and distinct structural features of plant CG maintenance methyltransferase MET1

Amika Kikuchi [1], Atsuya Nishiyama [2], Yoshie Chiba[2], Makoto Nakanishi [2], Taiko Kim To [3] & Kyohei Arita [1] ✉

DNA methylation is essential for genomic function and transposable element silencing. In plants, DNA methylation occurs in CG, CHG, and CHH contexts (where H = A, T, or C), with the maintenance of CG methylation mediated by the DNA methyltransferase MET1. The molecular mechanism by which MET1 maintains CG methylation, however, remains unclear. Here, we report cryogenic electron microscopy structures of *Arabidopsis thaliana* MET1. We find that the methyltransferase domain of MET1 specifically methylates hemimethylated DNA in vitro. The structure of MET1 bound to hemimethylated DNA reveals the activation mechanism of MET1 resembling that of mammalian DNMT1. Curiously, the structure of apo-MET1 shows an autoinhibitory state distinct from that of DNMT1, where the RFTS2 domain and the connecting linker inhibit DNA binding. The autoinhibition of MET1 is relieved upon binding of a potential activator, ubiquitinated histone H3. Taken together, our structural analysis demonstrates both conserved and distinct molecular mechanisms regulating CG maintenance methylation in plant and animal DNA methyltransferases.

DNA methylation is an evolutionarily conserved epigenetic mark that plays an important role in genome functions and stability[1–4]. In mammals, DNA methylation occurs at the 5th carbon atom of cytosine in the context of CG dinucleotides, with non-CG methylation being restricted to some cell types such as embryonic stem cells and neuronal cells[5,6]. Mammalian DNA methylation is established by de novo DNA methyltransferases DNMT3A/3B in cooperation with a paralog DNMT3L and maintained by DNMT1 and its bona fide recruiter Ubiquitin like with PHD and RING finger domains 1 (UHRF1)[7–13]. In contrast, DNA methylation in plants occurs not only in the CG context but also in the CHG (H denotes A, T, or C) and CHH contexts[14–17]. De novo DNA methylation at CG, CHG, and CHH is established by RNA-directed DNA Methylation (RdDM), in which small RNAs trigger the recruitment of de novo DNA methyltransferase, DOMAINS REARRANGED METHYLTRANSFERASE 2 (DRM2)[16,18,19]. Maintenances of CG, CHG, and CHH methylation is accomplished by DNA METHYLTRANSFERASE 1 (MET1), CHROMO-METHYLASE 3 (CMT3), and CMT2, respectively, in which the latter two enzymes recognize dimethylation at K9 of histone H3 (H3K9me2) on the nucleosomes[14,15,20–26]. Unlike animals, plants do not possess the ability from one place to another, and therefore, evolutionarily acquire phenotypes that are suitable for adapting to environmental changes, temperature, and drought. Such phenotypes are, in part, mediated by epigenetic modifications such as DNA methylation[27–29], suggesting that the inheritance of DNA methylation patterns may underpin plant functions and survival in a transgenerational manner.

The molecular mechanism of CG methylation maintenance in plants is believed to be analogous to that in mammalian systems. Plants have CG maintenance methyltransferase MET1 and its potential recruiters VARIANT IN METHYLATION 1 (VIM1), VIM2, and VIM3. The VIM proteins are methyl-DNA binding protein and ubiquitin E3-ligases

[1]Structural Biology Laboratory, Graduate School of Medical Life Science, Yokohama City University, Yokohama, Kanagawa, Japan. [2]Division of Cancer Cell Biology, The Institute of Medical Science, The University of Tokyo, Tokyo, Japan. [3]Life Science and Technology, Institute of Science Tokyo, Yokohama, Kanagawa, Japan. ✉e-mail: aritak@yokohama-cu.ac.jp

and are also the counterparts of the mammalian UHRF1 protein[30–36]. MET1 and VIM proteins are essential for CG maintenance methylation in transposable elements and protein-coding genes[14,37–39]. In mammals, a SET and RING-associated (SRA) domain of UHRF1 specifically recognizes hemimethylated DNA[40–42] and UHRF1 subsequently ubiquitinates histone H3 and PCNA-associated factor 15 (PAF15) with multiple mono-ubiquitination state depending on the replication timing[43–48]. The ubiquitinated proteins recruit DNMT1 to hemimethylation sites and trigger DNMT1 activation[44,49–52].

MET1 possesses replication foci target sequence (RFTS) 1–2, acidic-linker, bromo adjacent homology (BAH) 1–2 and methyltransferase (MTase) domains, whereas DNMT1 possesses PCNA-binding[53,54], RFTS, CXXC, autoinhibitory linker, BAH1-2 and MTase domains (Fig. 1a). Of note, MET1 has two RFTS domains and lacks the CXXC domain. Previous structural studies of various states of DNMT1 have revealed that apo-DNMT1 adopts an autoinhibitory conformation, in which the DNA-binding site of the MTase domain is covered by the RFTS domain and the autoinhibitory linker[55,56]. Multiple mono-ubiquitinated H3 catalyzed by UHRF1 binds to the RFTS domain, relieving autoinhibition and stimulating DNA methylation activity of DNMT1[50]. The activation state of DNMT1 is further characterized by a straight conformation of the DNA Recognition Helix in the MTase domain and helix-turn conformation of the Activating Helix, in which

Phe631 and Phe632 of the Activating Helix are inserted into activation/inactivation regulatory hydrophobic pocket in the MTase domain (Toggle Pocket)[49]. The CXXC domain of DNMT1 binds to unmethylated CG sequence and is involved in eliminating the unmethylated DNA from the MTase domain, resulting in suppression of de novo DNA methylation activity of DNMT1[57]. Although structural and functional features of plant MET1 are analogous to mammalian DNMT1, a number of questions still remain unclear, for example: how MET1 exerts preference methylation for hemimethylated CG DNA despite lacking the CXXC domain? Also, what are the architectures of activation and inactivation states of MET1? If MET1 activity is regulated by autoinhibitory mechanisms like DNMT1, what factors stimulate the methylation activity of MET1 and relieve the autoinhibitory state?

In this study, to address the questions raised above, we aimed to reveal conserved and distinct structural features for the CG maintenance methylation by *Arabidopsis thaliana* MET1. Cryogenic electron microscopy (cryo-EM) single-particle analysis of MET1 MTase domain bound to hemimethylated DNA revealed that the molecular features of active MET1 were similar to that of mammalian DNMT1. Cryo-EM structure of full-length apo-MET1 showed the autoinhibition mechanism to prevent DNA-binding of MET1 were distinct from that of DNMT1, indicating a plant-specific autoinhibitory mechanism. Furthermore, biochemical assays indicated that ubiquitinated H3 func-

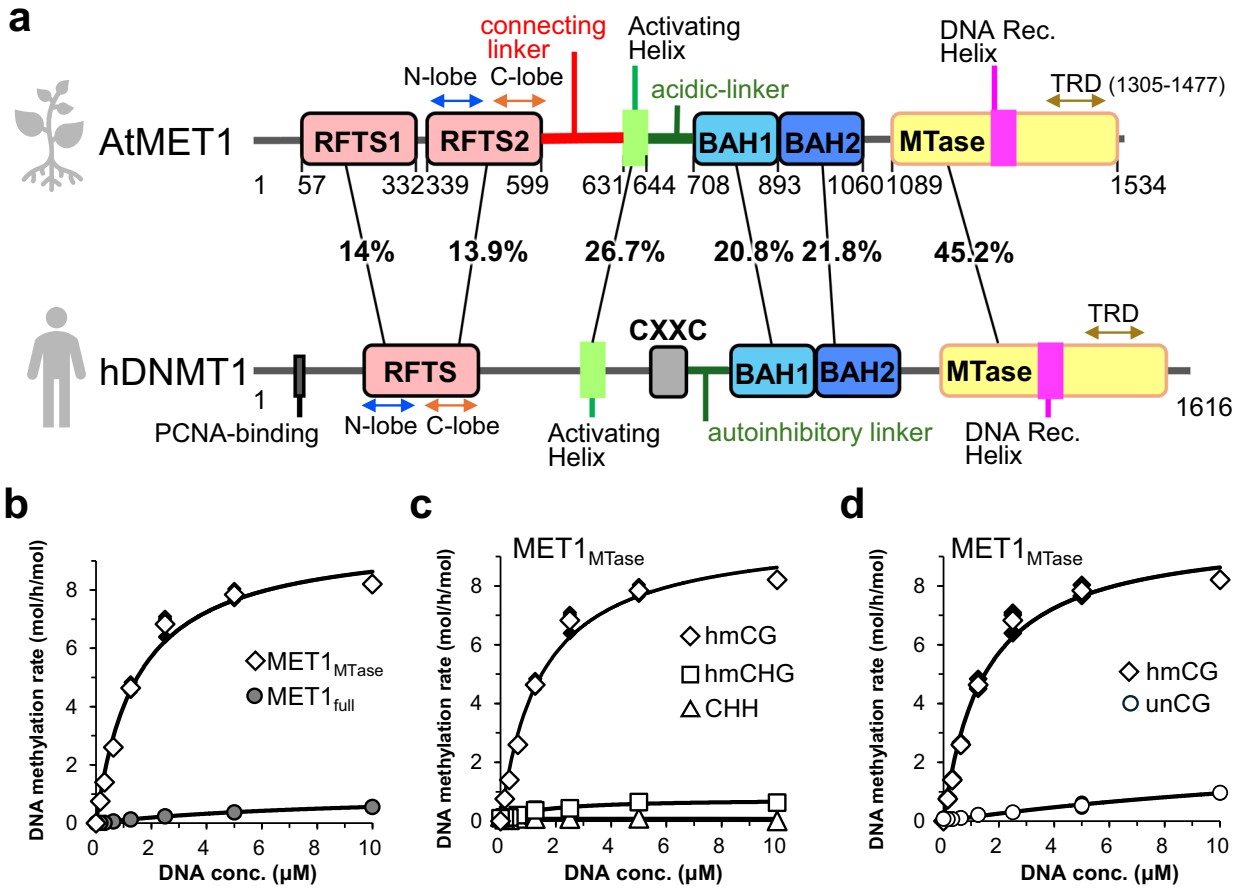

**Fig. 1 | Substrate specificity of MET1. a** Domain structures of *Arabidopsis thaliana* MET1 (AtMET1, top) and *Homo sapiens* DNMT1 (hDNMT1, bottom). Amino acid numbers and sequence identities are indicated. **b** In vitro DNA methylation assay of MET1$_{MTase}$ (white diamonds) and MET1$_{full}$ (gray circles) against hemimethylated DNA. The vertical axis indicates the turnover frequency of the methylation reaction. Michaelis–Menten curves are shown as lines. Data are presented as mean and standard deviation (SD) for $n$ = 3 independent biological replicates. **c** In vitro DNA

methylation assay of MET1$_{MTase}$ for hmCG (white diamonds), hmCHG (white squares), or CHH (white triangles) DNA. Data are presented as the mean ± SD for $n$ = 3 independent biological replicates. **d** In vitro DNA methylation assay of MET1$_{MTase}$ on a DNA duplex containing hemimethylated (white diamonds) and unmethylated (white circles) CG sites. Data are presented as the mean ± SD for $n$ = 3 independent biological replicates.

tions as a potential activator of MET1 to relieve the autoinhibitory state. Our structural data, therefore, revealed both autoinhibition and activation mechanisms of MET1, and provided structural implications for the maintenance of MET1-mediated DNA methylation in CG contexts in plants.

## Results

### Enzymatic activity and substrate specificity of *Arabidopsis thaliana* MET1

To evaluate the enzymatic activity and substrate specificity of *Arabidopsis thaliana* MET1, we conducted an in vitro DNA methylation assay. This assay used full-length MET1 (MET1$_{full}$) and a version of MET1 lacking the RFTS1–2 domains (MET1$_{MTase}$, aa: 621–1534, Fig. 1a), along with a 42-base pair DNA duplex containing CG dinucleotides in hemimethylated or unmethylated states (Supplementary Fig. 1a). The most effective DNA methylation activity of both MET1$_{full}$ and MET1$_{MTase}$ was exerted at 30 °C, which was supported by the results of a thermal stability assay showing that the melting temperatures ($T_m$) of MET1$_{full}$ and MET1$_{MTase}$ were 37.4 and 35.4 °C, respectively, which are lower than DNMT1 (Supplementary Fig. 1b, c). The DNA methylation activity of MET1$_{full}$ was ~15-fold lower than MET1$_{MTase}$, strongly indicating that RFTS1 and/or 2 domains in the N-terminal region play an autoinhibitory role (Fig. 1b). MET1$_{MTase}$ showed methylation activity preferable to CG context than CHG or CHH contexts containing hemimethylation sites (Fig. 1c and Supplementary Fig. 1a), in line with its dominant role of MET1 in CG methylation[58]. Furthermore, MET1$_{MTase}$ showed more than 9-fold higher enzymatic activity to hemimethylated CG DNA than unmethylated CG DNA (Fig. 1d), indicating preference for hemimethylated CG DNA although MET1 intrinsically lacks the CXXC domain. These data suggested that the MTase domain of MET1 has a regulatory mechanism for selective methylation of hemimethylated CG DNA.

### Cryo-EM structure of MET1$_{MTase}$ bound to hemimethylated DNA

To uncover the activation state of MET1, we determined the cryo-EM structure of MET1$_{MTase}$ bound to 12 bp hemimethylated DNA in which the target cytosine in hemimethylation site was replaced by a 5-fluorocytosine (5fC) to form an irreversible covalent complex with MET1 (Fig. 2a)[59]. MET1$_{MTase}$ was used to prepare the complex, because in vitro DNA methylation assay implied the autoinhibitory state of MET1$_{full}$ (Fig. 1b). The MET1$_{MTase}$ bound to the DNA was purified with size-exclusion chromatography (Supplementary Fig. 2) and subjected to cryo-EM (Supplementary Fig. 3 and Supplementary Table 1). Cryo-EM map of the MET1$_{MTase}$ bound to hemimethylated DNA complex was successfully reconstructed at 2.74 Å resolution, which enables us to generate the atomic model of the complex, except for the acidic-linker (aa: 646–707), loops and linkers in BAH1 (aa: 748–751), and BAH2 (aa: 925–933, 958–965, 1050–1054 and 1059–1089) (Figs. 1a, 2b, and Supplementary Fig. 3). Domain assembly of the BAH1, BAH2 and MTase domains of MET1 was similar to that of DNMT1, with root mean square deviation (RMSD) of 0.768 Å over 682 aligned Cα atoms, indicating the conserved rigid core structure (Fig. 3a). The MTase domain of MET1 consists of two moieties, catalytic core containing S-adenosyl homocysteine (SAH, a methyl donor cofactor product) binding site and target recognition domain (TRD: 1305–1477) (Figs. 1a, 2b). Hemimethylated DNA was sandwiched by the catalytic core and TRD, in which TRD loop1 (aa:1426-1434) interacted with the backbone phosphates of the non-target strand of the DNA (Fig. 2c and Supplementary Fig. 3g). In addition, the main chain carbonyl of Met1461 and side chain of Lys1463 in TRD loop2 (aa: 1458–1470) interacted with 5-methylcytosine (5mC) and the adjacent guanin in 5mCG of the non-target strand, respectively (Fig. 2d and Supplementary Fig. 3g). The methyl group of 5mC in the non-target strand was tightly surrounded by Cys1427, Leu1428, Trp1438, Leu1441, Met1461 and Gly1462 of the TRD via van der Waals interactions (Fig. 2e and Supplementary Fig. 3g).

Flipped-out 5fC in the target strand was covalently bound to sulfhydryl group of Cys1198 of the catalytic loop in MET1 (aa: 1196–1210), and recognized by side chains of Glu1238, Arg1282 and Arg1284 of the catalytic core (Fig. 2f and Supplementary Fig. 3g), in which side chain of Met1204 occupies the vacant space after base flipping of 5fC (Fig. 2g and Supplementary Fig. 3g). The spatial positions of the 5fC and 5mC bases in the hemimethylated DNA were strictly restricted by the catalytic loop and TRD, respectively, which ensure the robust CG sequence methylation of MET1[60]. The amino acid residues responsible for recognizing 5mCG and 5fC in *Arabidopsis thaliana* MET1 are highly conserved among MET1 orthologs in other plants, suggesting that plant MET1 proteins specifically recognize hemimethylated CG sites (Supplementary Fig. 4a).

In the N-terminal region of MET1, the Activating Helix (aa: 631–644) was adpted to a helix-turn conformation, which is similar to the conformation of the Activating Helix in the activation state of DNMT1[49] (Supplementary Fig. 5a). This conformation of the Activating Helix allowed the Arg1206 and Phe1207 side chains in the catalytic loop of MET1 to reach the minor groove at the 5mCG/5fCG site (Fig. 2h).

Taken together, the structural features of MET1 bound to hemimethylated DNA are totally similar to the activated form of DNMT1 bound to DNA, indicating the structural conservation of the activation state between mammalian and plant CG maintenance DNA methyltransferases.

### Structural comparison of MET1 with DNMT1

To illustrate the structural features that are specific to MET1 and those conserved among mammalian and plant CG maintenance DNA methyltransferases, we compared the hemimethylated DNA-binding structures of MET1 and DNMT1 (Fig. 3a and Supplementary Fig. 6). Significant structural differences have been observed in the TRD of the MTase domain[61]. For example, DNMT1 has a large insertion sequence in the TRD, with Cys1476-Cys1478-Cys1485-His1502 residues coordinating to the zinc atom (Fig. 3a left and Supplementary Fig. 6). In addition, the TRD of DNMT1 interacts with an α-helix (aa: 963-979) in a loop between BAH2 and MTase (hereafter BAH2-loop), forming a hydrogen bond between Tyr976 in the α-helix and His1509 of TRD, which presumably contributes to the specific binding to hemimethylated DNA (Fig. 3a right and Supplementary Fig. 6)[60]. In contrast, TRD of MET1 lacks the zinc-binding motif and is more compact than that of DNMT1 (Fig. 3a left and Supplementary Fig. 6). Furthermore, because the BAH2-loop of MET1 was shorter than DNMT1, the TRD of MET1 was not supported by any other MET1 moiety, indicating that plant CG methyltransferases recognize hemimethylation sites via a compact TRD structure without further structural support (Fig. 3a right and Supplementary Fig. 6).

As the amino acid residues involved in DNA methylation are conserved between human DNMT1 and *Arabidopsis thaliana* MET1 (Supplementary Fig. 7), we selected and mutated candidate residues based on a previous mutational study performed on DNMT1[60], and evaluated the effect of these mutations by in vitro DNA methylation assay. This assay demonstrated that mutation in Trp1438 (W1438A) of MET1, which is involved in recognition of 5mC in the non-target strand, reduced DNA methylation activity (Fig. 3b). This is consistent with findings in mammalian DNMT1, where mutation of the corresponding residue, Trp1512 in mice, also abolished methylation activity[60]. Alanine mutations in Met1204, Arg1206, and Phe1207 in the catalytic loop also impaired DNA methylation activity of MET1 (Fig. 3b). The negative effects of these mutations in MET1 are concordant with those of DNMT1[60], indicating the functional and structural conservation of the catalytic and DNA recognition residues in human DNMT1 and *Arabidopsis thaliana* MET1. The structural feature of the activating state of DNMT1 is characterized by the insertion of side chains of Phe631/Phe632 in the Activating Helix into the Toggle Pocket (Supplementary

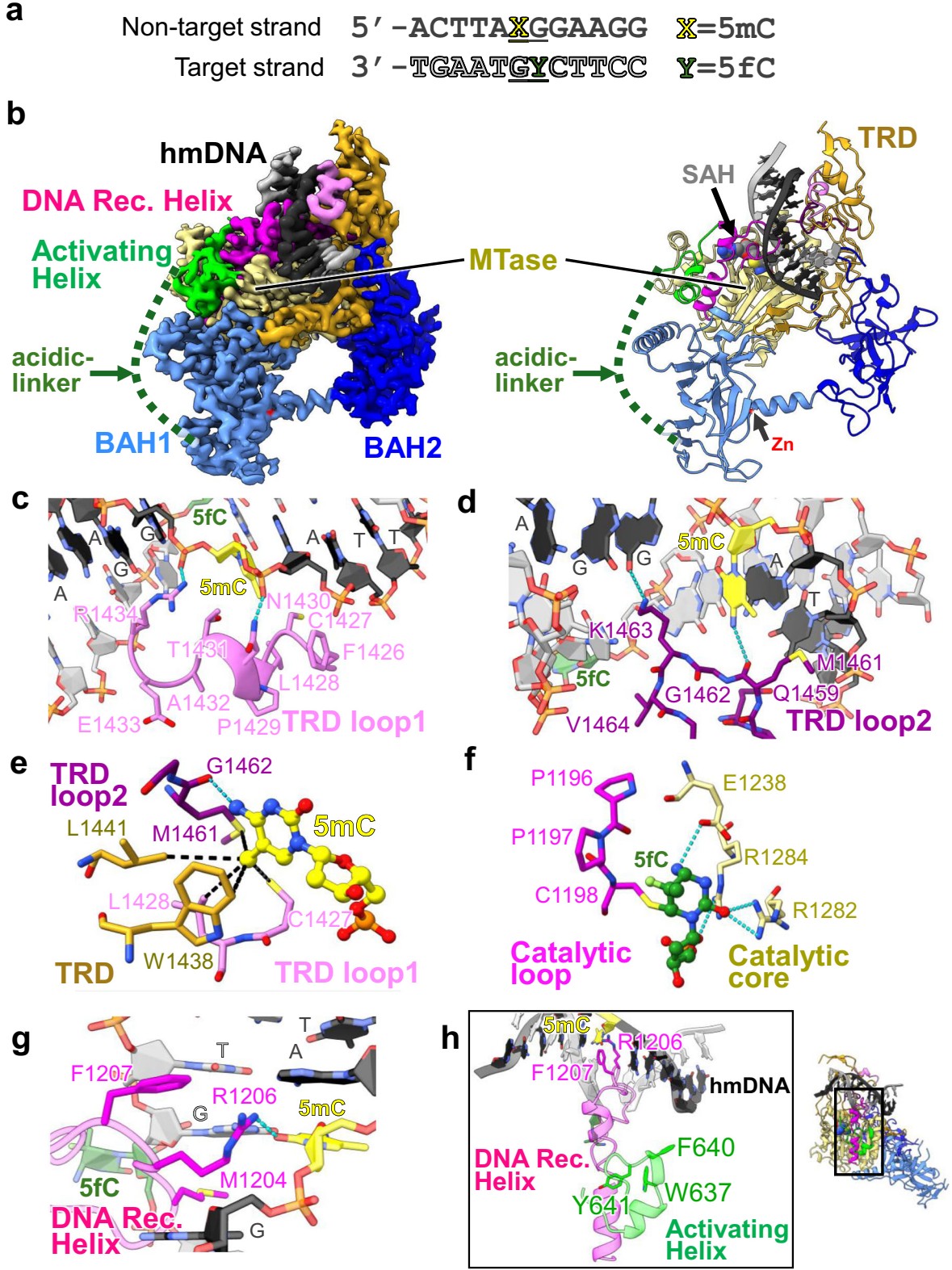

**a**

Non-target strand  5'–ACTTA**X**GGAAGG   **X**=5mC
Target strand     3'–TGAAT**GY**CTTCC   **Y**=5fC

Fig. 5a right). In contrast, the corresponding residues of MET1, Phe640 and Tyr641 in the Activating Helix, did not enter the Toggle Pocket (Supplementary Fig. 5a left). In fact, dual alanine mutations of Phe640 and Tyr641 of MET1 slightly affected DNA methylation activity (Fig. 3c), indicating a different role for aromatic residues in MET1 activation. However, the mutation of Trp637, which functions as an anchor of the Activating Helix to the MTase domain, to alanine decreased DNA methylation activity by 2-fold, revealing that proper positioning of the

Activating Helix is important for the DNA methylation activity of MET1 (Fig. 3c).

A previous genetic study has reported that two missense mutations in the MTase domain of MET1, G1101S (*met1-2*) and P1300S (*met1-1*), are associated with genome-wide DNA hypomethylation and phenotypic changes[58]. Cryo-EM analysis of MET1 revealed that Gly1101 is located close to SAH, indicating that the G1101S mutation causes steric clashes with SAH, leading to reduced DNA methylation activity

**Fig. 2 | Cryo-EM structure of the MET1$_{MTase}$ bound to hemimethylated DNA.**
**a** DNA sequences used for cryo-EM structural analysis. X and Y mean
5-methylcytosine (5mC) and 5-fluorocytosine (5fC), respectively. Underlined text
indicates the CG site. **b** Cryo-EM density map (left) and atomic model (right) of the
MET1$_{MTase}$ bound to hemimethylated DNA. The disordered acidic-linker is indicated
by the green dotted lines. The zinc ion and SAH molecule are shown as sphere
representations. **c** Interaction between TRD loop1 (light pink) and hmDNA (target
strand: gray, non-target strand: black). The hydrogen bonds are indicated by the
cyan dashed lines. The 5mC and 5fC are shown as yellow and green sticks,
respectively. **d** Recognition of hmDNA by TRD loop2 (purple). The hydrogen bonds
are indicated by the cyan dashed lines. The color schemes are the same as those in

Fig. 2c. **e** 5mC recognition by the TRD loop1 (light pink), TRD loop2 (purple), and
TRD (orange) of MET1. The black dashed line indicates van der Waals interactions.
**f** Recognition of 5fC. The 5fC, catalytic loop, and catalytic core are shown as green,
magenta, and yellow sticks, respectively. Cyan dashed line indicates hydrogen
bond. **g** Structure of the catalytic loop of MET1 for the recognition of DNA from the
minor groove at the 5mCG/5fCG site. The catalytic loop is shown as a magenta stick
model. **h** Structure around Activating Helix (green) and DNA Recognition Helix
(magenta). The left panel shows a magnified image. Residues of the Activating Helix
and DNA Recognition Helix are shown as green and magenta stick models,
respectively.

(Fig. 3d). P1300S mutation presumably causes steric hindrance with
the surrounding hydrophobic residues, indicating a negative effect on
folding of the MTase domain (Fig. 3e). These amino acid residues are
broadly conserved in methyltransferases from prokaryotes to
eukaryotes[57,58], suggesting their conserved involvement in methyl-
transferase activity.

### Autoinhibitory structure of apo-MET1$_{full}$
Having elucidated the activated form of MET1, we next turned our
attention to understanding its autoinhibitory state through the struc-
ture of apo-MET1$_{full}$ (Fig. 4a and Supplementary Fig. 8, and Supple-
mentary Table 1). Cryo-EM map of apo-MET1$_{full}$ was reconstructed at
3.21 Å resolution; the RFTS1 domain (aa: 1–345), the acidic-linker (aa:
648–707), loops in the BAH2 domain (aa: 926–934 and 958–965) and
the catalytic loop in the MTase domain were completely disordered
reflecting a highly flexible state (Fig. 4a). The domain assembly of the
core region of apo-MET1$_{full}$, BAH1–BAH2–MTase, was identical to that
of MET1$_{MTase}$ bound to the hemimethylated DNA complex with an
RMSD of 0.550 Å over 776 aligned Cα atoms (Fig. 4b). The RFTS2
domain of MET1 was composed of a β-barrel lobe (N-lobe, aa: 346–494)
and an α-helical bundle (C-lobe, aa: 495–590) (Fig. 1a). The structure of
apo-MET1$_{full}$ revealed that the RFTS2 domain contacted the TRD in the
MTase domain and residues 601–610 in the connecting linker (aa:
599–623) between the RFTS2 domain and the Activating Helix, con-
tributing to the stable spatial position of the RFTS2 domain (Fig. 4a, e
left). This spatial positioning of the RFTS2 domain caused a severe
steric clash with hemimethylated DNA, in which residues 360–366 and
397–405 of the N-lobe in the RFTS2 domain played a potential inhi-
bitory role in DNA-binding when compared to the DNA bound form of
MET1 (Fig. 4b, c). Intriguingly, the connecting linker was rigidly posi-
tioned across the axis of the DNA duplex bound to the catalytic core
(Figs. 1a, 4d), suggesting that the connecting linker reinforces the
autoinhibitory state of apo-MET1$_{full}$. Zooming in the catalytic site
revealed the structural differences of the DNA Recognition Helix and
the Activating Helix between apo-MET1$_{full}$ and MET1$_{MTase}$ bound to the
DNA. In the structure of apo-MET1$_{full}$, the DNA Recognition Helix
(aa:1208–1231) in the MTase domain was kinked at Met1219, resulting
in disorder of the subsequent catalytic loop, in which the Toggle
Pocket composed of Ile1220, Phe1235, Phe1246, Thr1251, and Leu1254
of the MTase domain accepted side chain of Val1215 of the DNA
Recognition Helix (Fig. 4e right and Supplementary Fig. 5b left). These
structural features differ from the corresponding region of the
MET1$_{MTase}$ bound to DNA, as MET1 exhibits a straight conformation of
the DNA Recognition Helix, and an ordered catalytic loop (Supple-
mentary Fig. 5a). The Activating Helix of apo-MET1$_{full}$ formed a three-
turn helical conformation, which differed from the turn conformation
following the two-turn helix of MET1$_{MTase}$ bound to the DNA complex
(Fig. 4e right and Supplementary Fig. 5). These structural features of
the DNA Recognition Helix and Activating Helix in apo-MET1$_{full}$ were
almost identical to those of the autoinhibitory state of DNMT1 (Sup-
plementary Fig. 5b).

Collectively, cryo-EM analysis revealed that the RFTS2 domain and
the connecting linker play a potential inhibitory role in

hemimethylated DNA-binding to the MTase domain in the auto-
inhibitory state of apo-MET1$_{full}$.

### Structural differences in autoinhibitory mechanism
Structural comparison between apo-MET1$_{full}$ and apo-DNMT1 (PDB ID:
4WXX) revealed significant differences in the spatial arrangement of
the RFTS domain (Fig. 5a, b). In apo-DNMT1, the C-lobe of RFTS domain
is deeply inserted into the DNA-binding pocket of the MTase domain
and inhibits the binding of hemimethylated DNA, which requires the
acidic residues of the C-lobe in RFTS domain to make hydrogen bonds
and ionic interactions with the MTase domain (Fig. 5a, c)[55]. In addition,
the basic residues of the RFTS domain donate hydrogen bonds to the
acidic residues in the autoinhibitory linker (Fig. 5c). In clear contrast,
the RFTS2 domain of MET1 was positioned in an upside-down orien-
tation and the N-lobe of the RFTS2 domain was associated with the
TRD in the MTase domain away from the DNA binding pocket (Fig. 5a,
d). Asp402, Asp420, and Asn422 of the N-lobe in the RFTS2 domain
formed hydrogen bonds with Arg1393, Lys1412, and Val1420 of TRD,
respectively, resulting in spatially different position from that of the
RFTS domain in DNMT1 (Fig. 5d).

It has been shown that the autoinhibitory linker of DNMT1 inserts
into the catalytic cleft of the MTase domain, contributing to the
autoinhibitory state of apo-DNMT1 (Fig. 5a, c)[55]. In MET1, the corre-
sponding region contains an acidic-linker (Figs. 1a, 5a). However, the
acidic-linker in MET1 is highly flexible, suggesting a limited auto-
inhibitory role (Figs. 4a, 5a, and 5c). Of note, MET1 possesses a specific
connecting linker between the RFTS2 domain and the Activating Helix,
whereas the corresponding region of DNMT1 is unstructured (Fig. 5a).
This connecting linker of MET1 presumably inhibits hemimethylated
DNA-binding due to severe steric clashes with the DNA (Fig. 4d).
Although the connecting linker sequence is poorly conserved among
plants, except for the NLNPxA motif (Fig. 4d and Supplementary
Fig. 4b), the linker length is conserved to within one residue. This
observation suggests that the physical length of the linker, rather than
its specific sequence, is essential for inhibiting DNA binding. There-
fore, while the function of autoinhibition is conserved, the molecular
mechanisms underlying CG maintenance methyltransferase auto-
inhibition differ between mammals and plants.

### Role of RFTS domains in MET1 activation
Next, we compared the structures of MET1-RFTS2 and the DNMT1-
RFTS domains (Fig. 6a). The N-lobe of DNMT1-RFTS domain has a
ubiquitin interacting motif (UIM) for recognition of the K18-linked
ubiquitin (K18ub) on histone H3, and a ubiquitin recognition loop
(URL) to separate K18ub and K23ub on histone H3[50,62]. The MET1 RFTS2
domain has a UIM-like motif from residues 368 to 383, in which
Glu369, Glu370, and Ser380 of RFTS2 domain are conserved (Fig. 6b)
and also has a conserved β-strand structure to form an intermolecular
interaction with the histone H3 tail (Fig. 6a). As ubiquitinated H3 is
catalyzed by UHRF1 in mammals, we examined the ubiquitination
activity of the UHRF1 plant homolog VIM1. In vitro ubiquitination
assays using mouse UBA1 (E1), *Arabidopsis thaliana* UBC11 (E2), and
VIM1 demonstrated that that VIM1 promoted ubiquitination of the

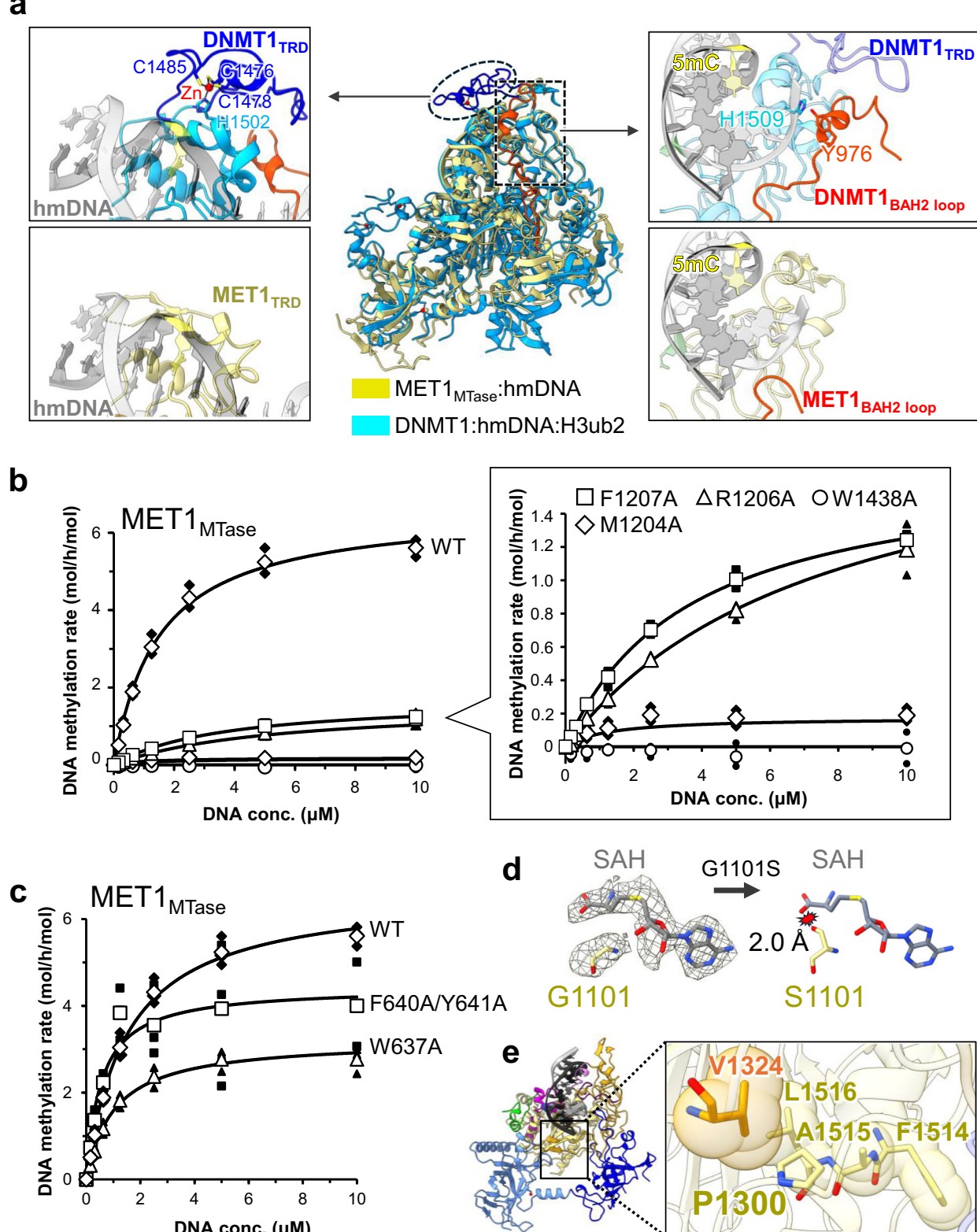

**Fig. 3 | Structural comparison of DNA bound forms of MET1 and DNMT1.**
**a** Superimposition of the structure of MET1$_{MTase}$ bound to hmDNA (yellow) on that of DNMT1 bound to hmDNA and H3ub2 (cyan, PDB: 7XI9). The magnified figures on the left show the TRD structures in which the upper and lower panels are DNMT1 and MET1, respectively. The magnified figures on the right display the structures of the BAH2 loop of DNMT1 (upper panel) and MET1 (lower panel). **b** DNA methyltransferase activity of MET1 mutants in the MTase domain. Data are presented as the mean ± SD for $n$ = 3 independent biological replicates. **c** DNA

methyltransferase activity of MET1 mutants in Activating Helix. Data are presented as the mean ± SD for $n$ = 3 independent biological replicates. **d** Structural model of G1101S mutation. SAH and Gly1101 in the left panel are depicted as gray and yellow sticks superimposed on the cryo-EM map, respectively. The right panel presents the Ser1101 substitution model. **e** Structure around P1300 of the MTase domain of MET1. P1300 and the residues of catalytic core are shown as yellow and orange stick transparent sphere, respectively.

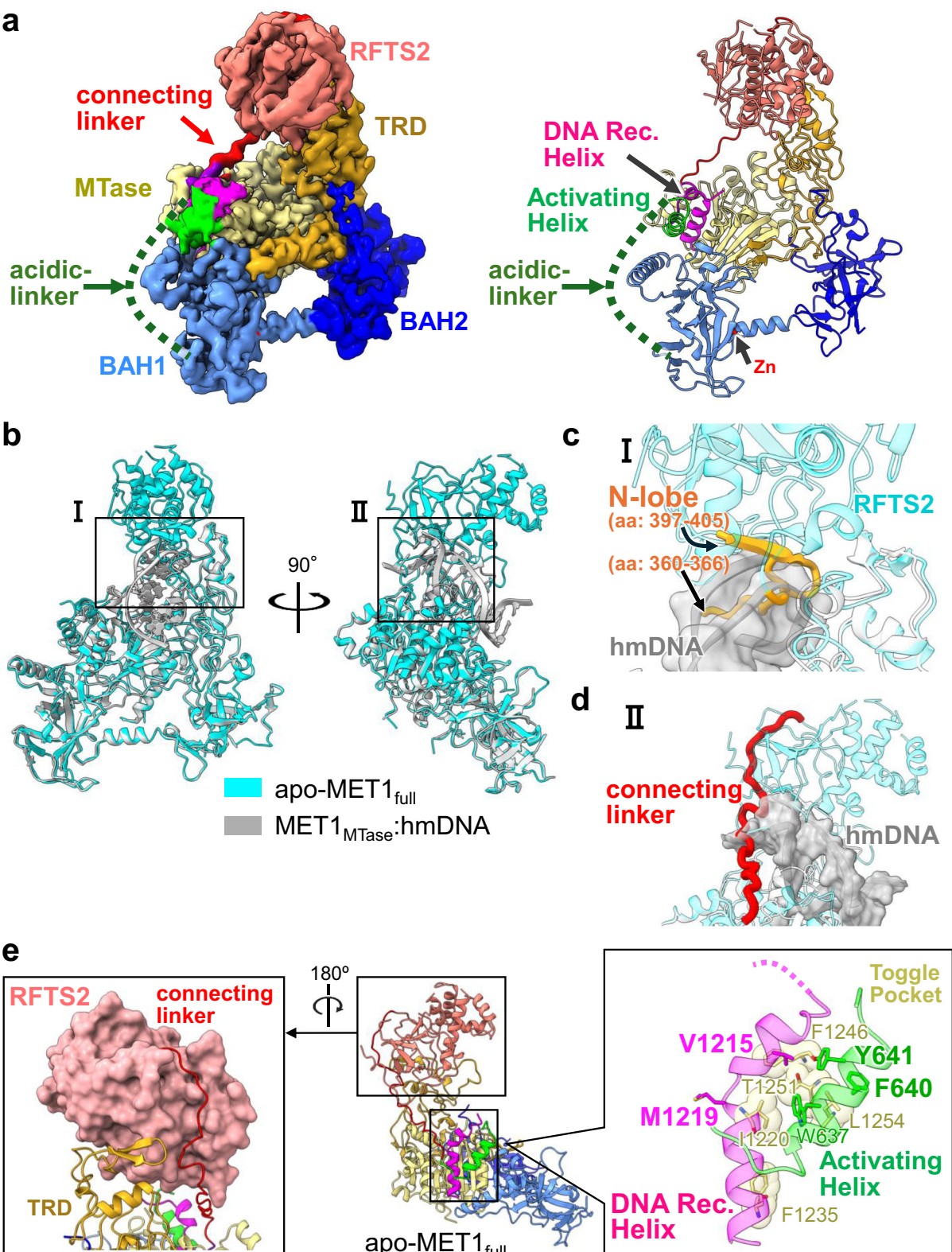

**Fig. 4 | Cryo-EM structure of apo-MET1_full. a** Cryo-EM density map (left) and cartoon model (right) of apo-MET1_full. Zinc ion is shown as red sphere. The disordered acidic-linker is indicated by the green dotted lines. **b** Structural alignment between apo-MET1_full (cyan) and MET1_MTase bound to hmDNA (gray). **c** A close-up view of the RFTS2 domain and hmDNA in Fig. 4b. hmDNA is shown as a gray stick model overlaying the cryo-EM map. The sterically hindered region of the N-lobe of the RFTS2 domain is shown in orange. **d** A close-up view of the connecting linker

(red) and hmDNA (gray) in Fig. 4b. **e** The left panel displays the structure of the RFTS2 domain, depicted as the salmon surface, bound to the connecting linker (red) and the TRD (orange). The right panel shows the structure around the Toggle Pocket (light yellow) of apo-MET1_full. The Toggle Pocket is shown as a stick model with a transparent sphere model. Residues in the Activating Helix and DNA Recognition Helix are shown as green and magenta stick models, respectively.

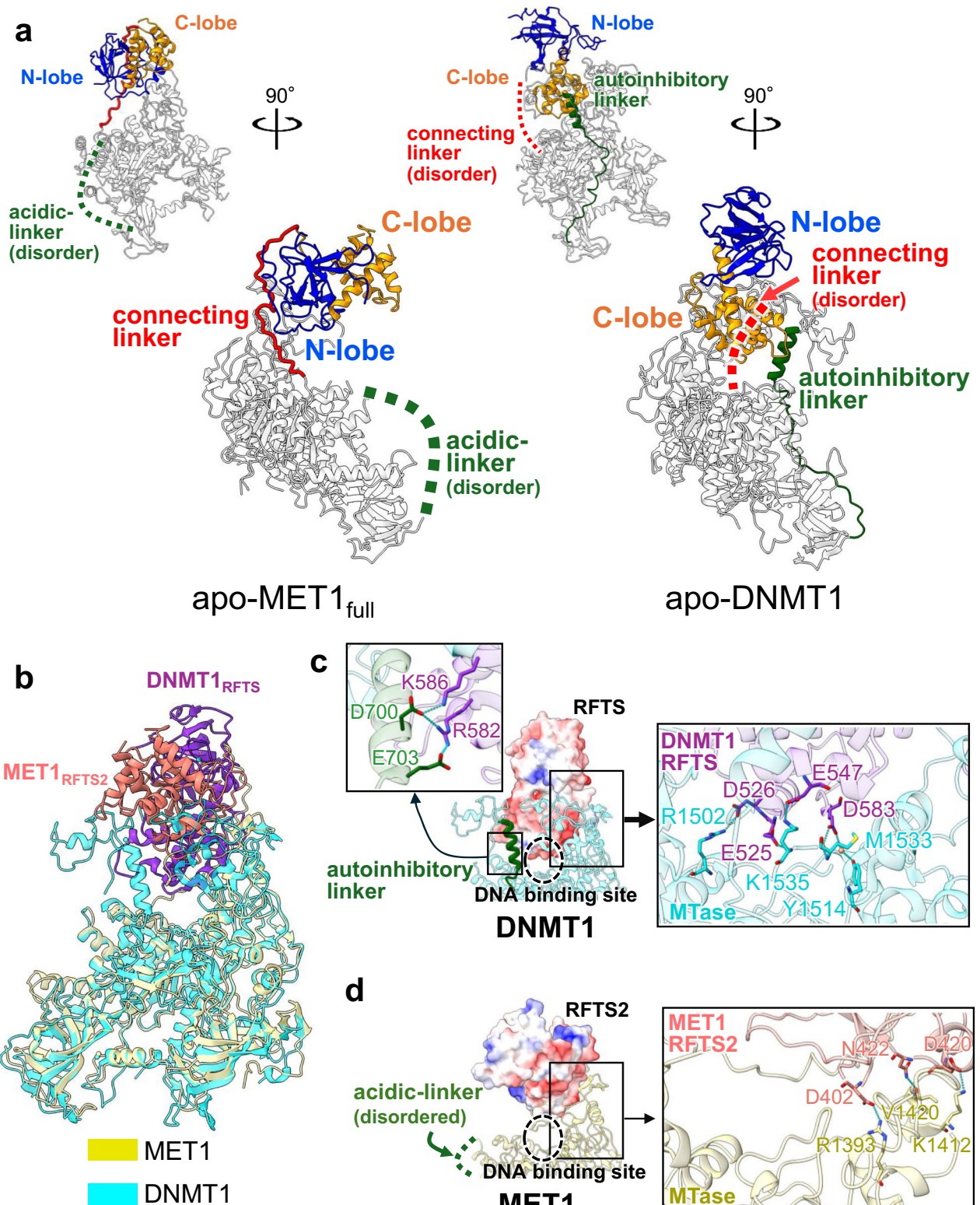

**Fig. 5 | Structural comparison of autoinhibitory states of MET1 and DNMT1.**
**a** Structures of apo-MET1$_{full}$ (left) and apo-DNMT1 (right, PDB: 4WXX). N-lobe, C-lobe, connecting linker, and autoinhibitory/acidic-linkers are colored blue, orange, red, and green, respectively. **b** Overlay of apo-MET1$_{full}$ (yellow) with apo-DNMT1 (cyan, PDB: 4WXX). The RFTS2 and RFTS domains are colored salmon and purple, respectively. **c** Close-up views on the left are the C-lobe of the RFTS domain (light purple) and the autoinhibitory linker (green), and the right shows the MTase domain of DNMT1 (cyan). The residues involved in the interactions are displayed as blue sticks. **d** Close-up view of the N-lobe of the RFTS2 domain (salmon) and the MTase domain (light yellow) in MET1.

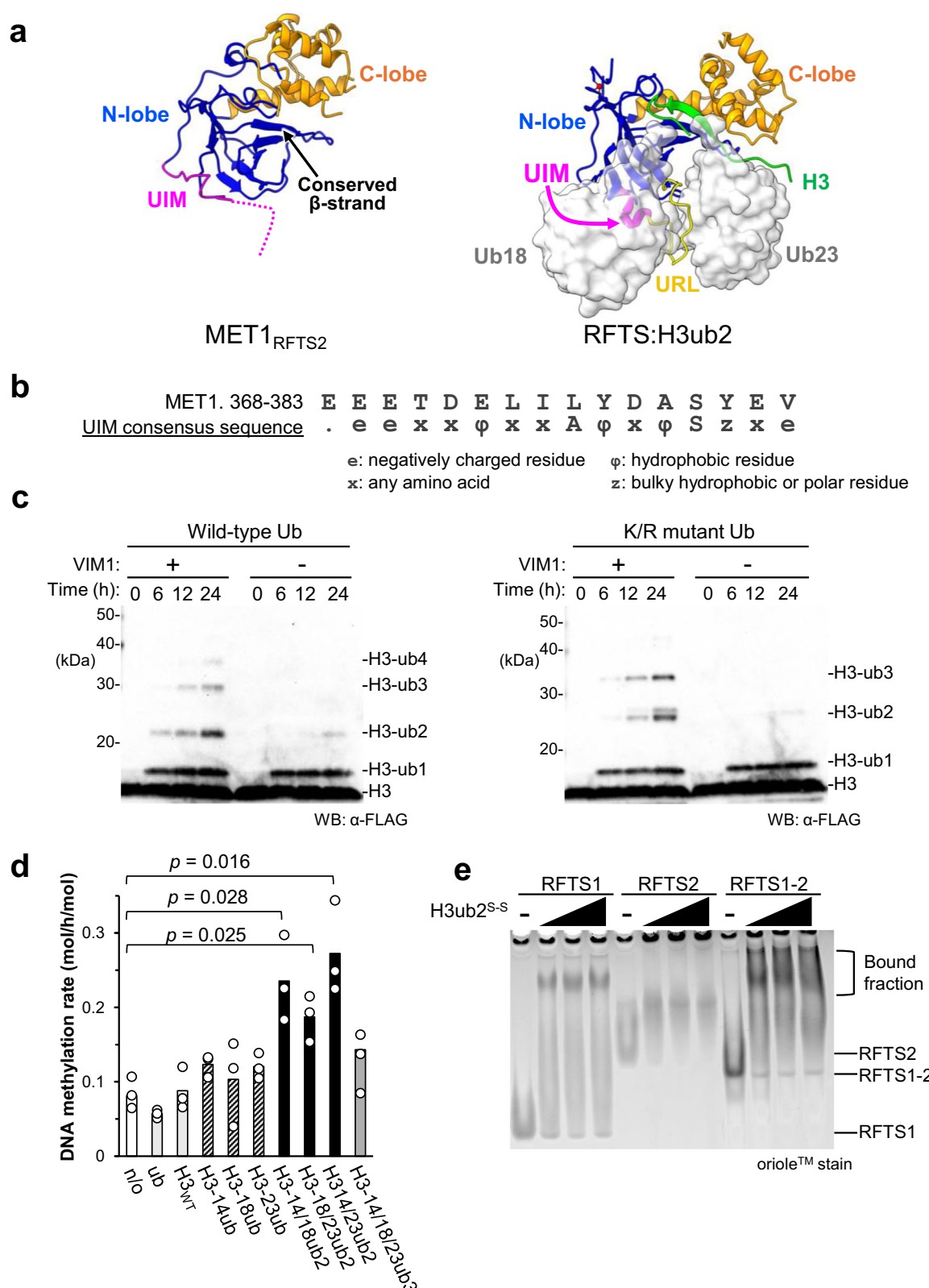

**b**

MET1. 368-383 E E E T D E L I L Y D A S Y E V

UIM consensus sequence . e e x x φ x x A φ x φ S z x e

e: negatively charged residue φ: hydrophobic residue
x: any amino acid z: bulky hydrophobic or polar residue

N-terminal tail of human histone H3.1 (aa: 1–37 W, $H3_{1-37W}$), a region fully conserved between human and plants (Fig. 6c and Supplementary Fig. 9a). Initial ubiquitination appeared to be non-specifically catalyzed by the E2 enzyme, potentially due to suboptimal experimental conditions, such as the use of an E2 concentration higher than the physiological level. In vitro ubiquitination assay using a K/R ubiquitin mutant, in which all lysine residues were replaced with arginine

residues to prevent poly-ubiquitination, showed a ubiquitination pattern similar to that of wild-type ubiquitin, indicating that VIM1 catalyzes multiple mono-ubiquitination of histone H3 tail (Fig. 6c).

Based on these results, we hypothesize that ubiquitinated histone H3 tail, catalyzed by VIM proteins, engages in the release of MET1 autoinhibition by binding to the RFTS domains. To test the possibility, in vitro DNA methylation assays were performed in the presence of 50

**Fig. 6 | Activation of MET1 by ubiquitinated histone H3. a** Structural comparison of the MET1 RFTS2 domain (left) and the DNMT1 RFTS domain bound to ubiquitinated H3 (right, PDB: 5WVO). Color schemes for the RFTS domains are consistent with those in Fig. 5. In the right panel, the histone H3 tail and ubiquitins are shown as green cartoons and transparent surface models, respectively. **b** Amino acid sequence of the putative UIM motif in the RFTS2 domain of MET1. **c** In vitro ubiquitination of C-terminal FLAG tagged H3 tail by VIM1 using wild-type ubiquitin (left) or K/R ubiquitin mutant (right). The ubiquitinated H3 was detected using anti-FLAG antibody. Experiments were independently repeated three times with similar results. **d** In vitro DNA methylation assay of MET1$_{full}$ in the presence of ubiquitin

(ub), histone H3 tail (H3$_{WT}$), mono-ubiquitinated H3s: H3C14ub (H3-14ub), H3C18ub (H3-18ub), and H3C23ub (H3-23ub), dual mono-ubiquitinated H3s: H3C14ub/18ub (H3-14/18ub2), H3C18ub/C23ub (H3-18/23ub2), and H3C14ub/C23ub (H3-14/23ub2), and triple mono-ubiquitinated H3: H3C14ub/C18ub/C23ub (H3-14/18/23ub3). DNA methylation rates were monitored after 1 h incubation with 10 μM hemimethylated DNA at 30 °C. Data are presented as the mean ± SD for $n$ = 3 independent biological replicates. Statistical significance was assessed using a two-tailed Student's $t$-test relative to MET1$_{full}$. **e** Interaction between a series of RFTS domains of MET1 and the H3C18ub/C23ub analog (H3ub2$^{S-S}$) analyzed by EMSA. Experiments were independently repeated three times with similar results.

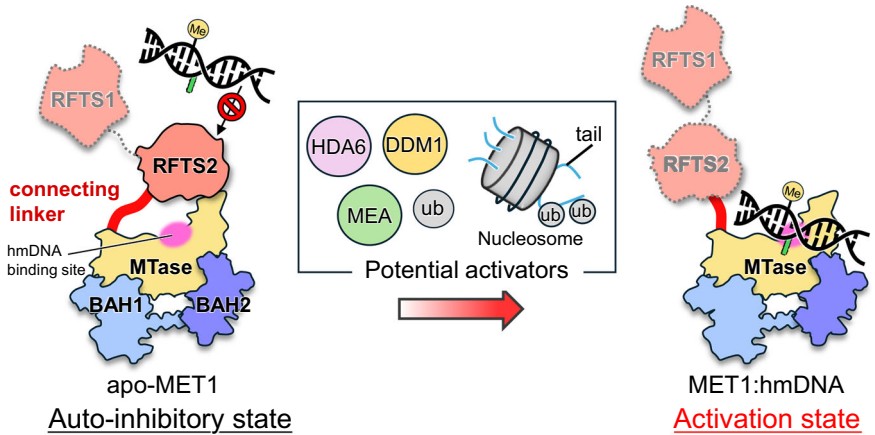

**Fig. 7 | Schematic representation of the MET1 activation mechanism.** The left panel illustrates the autoinhibitory state of MET1, in which the RFTS2 and connecting linker obstruct access of hmDNA to the catalytic center. The right panel shows the activated state of MET1, in which activation factors induce a spatial rearrangement of RFTS2 domain and connecting linker, allowing hmDNA to access the catalytic domain.

equimolar excess of the ubiquitinated H3$_{1-37W}$ analogs in which G76C ubiquitin was linked to K14C, K18C, K23C, K14C/K18C, K18C/K23C, K14C/K23C, and K14C/K18C/K23C (Fig. 6d and Supplementary Fig. 1e). These candidate ubiquitination patterns were designed based on known H3 ubiquitination sites mediated by UHRF1 in mammals[50]. This assay showed that the dual mono-ubiquitinated H3 analogs (H3C14ub/C18ub, H3C18ub/C23ub, and H3C14ub/C23ub) effectively enhanced DNA methylation activity of MET1 by 2-fold compared to that of apo-MET1, although the magnitude of activation was relatively modest (Fig. 6d). An electrophoretic mobility shift assay (EMSA) demonstrated that the dual mono-ubiquitinated H3 bound to either RFTS1, RFTS1−2, or RFTS2 domains of MET1 (Fig. 6e). These findings suggested that dual mono-ubiquitinated H3 functions as a potential activator of MET1 to relieve the autoinhibitory state.

## Discussion

Cryo-EM analyses of *Arabidopsis thaliana* CG maintenance methyltransferase MET1 provided structural insights into its activation and autoinhibitory states. The structure of the MET1 MTase domain bound to hemimethylated DNA showed a similar structural property to the DNA bound form of human DNMT1 characterized by recognition of 5mC in the non-target strand by TRD, recognition of flipped-out target base by the catalytic core and catalytic loop, and conformation of DNA Recognition Helix and Activating Helix are totally identical to those in DNMT1[49,60]. As expected from the previous sequence alignment analyses[61], MET1 has a compact TRD structure compared to DNMT1. In contrast, some differences were observed in the autoinhibitory state between apo-MET1 and apo-DNMT1. The spatial orientation of the RFTS2 domain of MET1 is upside-down manner compared to the RFTS domain of DNMT1, in which AlphaFold failed to predict the RFTS2 orientation of MET1 (AlphaFold Protein Structure Database: AF-P34881-F1-v4, Supplementary Fig. 10a)[63].

Despite these findings, however, several questions remain unanswered yet. Dual mono-ubiquitinated H3 (H3ub2) is a potential candidate for MET1 activation and indeed, H3ub2 increased DNMT1 methylation activity by 7-fold (Supplementary Fig. 1d). However, H3ub2 promoted the DNA methylation activity of MET1 by only 2-fold, which is significantly lower than that of MET1$_{MTase}$ (which lacks the RFTS1−2 domains, Figs. 1b and 6d). The relatively modest enhancement of MET1 activity by H3ub2 may be attributed to an insufficient amount of H3ub2 to fully saturate MET1 binding, as well as the use of an H3 tail peptide rather than a more physiologically relevant nucleosome substrate. These observations suggest that additional activation factors are required for the full activation of MET1 (Fig. 7). Consistent with this notion, mammalian DNMT1 binds the non-coding RNA/G-quadruplex[64,65] and *Neurospora Crassa* DNA methyltransferase Dim-2, an ortholog of DNMT1 and MET1, is activated by combined binding of H3K9me3 and heterochromatin protein 1[66]. The plant HISTONE DEACETYLASE 6 (HDA6) directly binds to MET1 and silence transposable element[67,68]. In addition, histone-lysine N-methyltransferase MEDEA (MEA) also directly interact with N-terminal region of MET1 in vitro, suggesting a possibility of functional modulation of the RFTS domains (Fig. 7)[69]. These findings suggest that histone modifications and other chromatin-associated biomolecules may regulate MET1 enzymatic activity.

At the same time, plant-specific activation mechanisms of MET1 are also possible. VIM proteins are genetically associated with MET1[34]. We examined the DNA methylation activity of MET1 in the presence of VIM1 but failed to observe an enhancement in the methylation activity of MET1 (data not shown). There are several possible scenarios for MET1 activation. A comprehensive MS/MS analysis that identified ubiquitinated proteins in *Arabidopsis thaliana* detected a Lys583 ubiquitinated peptide of MET2[70]. The ubiquitination site of MET2 is highly conserved in the RFTS2 domain of MET1 (K599 in the $^{598}$KKKK$^{601}$

cluster), and is also highly conserved in plants, but not in animals (Supplementary Fig. 10b). Interestingly, AlphaFold3 structural prediction[71] using the MET1 RFTS2 domain and ubiquitin as an input sequence suggests that ubiquitin is tightly bound to the RFTS2 domain, in which the C-terminus of ubiquitin is close to the N(ζ) atom of Lys599 in the lysine cluster, reflecting the formation of an isopeptide bond (Supplementary Fig. 10c). Although the ubiquitin E3 ligase for MET1 remains unidentified, it is possible that ubiquitination including, poly-ubiquitination, induces a structural change in the autoinhibitory state of MET1, leading to full activation. In this study, the role of the RFTS1 domain in autoinhibition remains unclear, as the cryo-EM map of RFTS1 domain in apo-MET1$_{full}$ was completely invisible (Fig. 4a). MET1 activation through the collaborative transaction of the two RFTS domains is also a possibility; however, this will be explored in future research.

It is not yet known which histones are the targets for ubiquitination by VIM proteins. Mammalian UHRF1 reportedly catalyzes the ubiquitination of histone H2B in addition to histone H3, which recruits DNMT1 to the replication sites during S phase[72]. A structural study of the plant homeodomain (PHD) finger of VIM1 has reported that the PHD finger does not show any binding ability to canonical histones H2A, H2B, H3, and H4[73], suggesting that histone variants or unidentified biomolecules have a possibility to undergo the VIM1-mediated ubiquitination. Interestingly, loss of the *met*1 gene severely impairs chromatin binding of the SWI/SNF2 chromatin remodeling factor DECREASE IN DNA METHYLATION 1 (DDM1), suggesting that MET1 physically colocalizes with DDM1[74]. DDM1 contributes to the deposition of H3.1 and H2A.W variants in nucleosomes for heterochromatin formation and transposons silencing, respectively[74–76], suggesting another scenario for MET1 activation mediated by ubiquitinated histone variants.

*Arabidopsis thaliana* MET1 exhibits a plant-specific structural feature in the autoinhibition state, and a conserved activation state compared to mammalian DNMT1. Although the function of autoinhibition is conserved among plants and animals, the molecular mechanism underlying the autoinhibition is markedly different. To the best of our knowledge, sequence comparisons suggest that plants are the only organisms with two RFTS domains (Supplementary Fig. 9b). Unlike animals, DNA methylation patterns in plants are inherited to the next generation without reprogramming. Therefore, the strict regulation of MET1 activity by the two RFTS domains might be essential for robust transgenerational inheritance of DNA methylation in plants. Further studies to identify the targets for VIM1-dependent ubiquitination and additional biomolecular factors for MET1 full activation are required to understand the robustness of plant DNA methylation inheritance.

## Methods

### Oligonucleotides
42 bases oligonucleotides, the sequences are shown in Supplementary Fig. 1a, for in vitro DNA methylation assay and twelve-base oligonucleotides (upper: 5′- ACTTA(5mC)GGAAGG, lower: 5′- CCTTC(5fC)GTAAGT) for cryo-EM single particle analysis were synthesized by GeneDesign, *Inc.* (Osaka, Japan). Oligonucleotides were dissolved in 10 mM HEPES (pH 7.5), and equimolar complementary strands were mixed, heated to 95 °C for 2 min, and annealed overnight at 4 °C to form DNA duplexes. Details of primers used for the construction of expression vectors and for site-directed mutagenesis are provided in Supplementary Table 2.

### Protein expression and purification
*Arabidopsis thaliana* full-length MET1 (MET1$_{full}$, UniProt: P34881) including the N-terminal ten histidine tag (His-tag) following the human rhinovirus 3 C (HRV 3 C) protease site, was expressed in Sf9 insect cells. Baculoviruses were produced using a BestBac v-cath/chiA

Deleted Baculovirus Co-transfection kit, according to the manufacturer's instructions. MET1$_{full}$ was expressed in Sf9 cells after incubation for 72 h at 27 °C. The cells were lysed using lysis buffer (20 mM Tris-HCl (pH 8.0) containing 300 mM NaCl, 5 mM imidazole, 10% (w/v) glycerol, and a protease inhibitor cocktail (Nacalai)) and sonicated. After centrifugation to remove debris, a soluble fraction was loaded onto TALON Superflow Metal Affinity Resin (Takara, Cat#635670), and unbound proteins were washed with wash buffer (20 mM Tris-HCl (pH 8.0) containing 1 M NaCl, 5 mM imidazole, 10% (w/v) glycerol), and lysis buffer. Bound proteins were eluted using elution buffer (50 mM Tris-HCl (pH 8.0) containing 500 mM imidazole, 300 mM NaCl, and 10% (w/v) glycerol). The His-tag was cleaved using the HRV3C protease at 4 °C for over 12 h. MET1$_{full}$ was separated by anion-exchange chromatography, HiTrap Heparin HP (Cytiva, Cat#17040701) using a gradient of 50 to 1000 mM NaCl in 20 mM Tris-HCl (pH 8.0) buffer containing 10% (w/v) glycerol and 0.5 mM tris (2-carboxyethyl) phosphine (TCEP). Finally, MET1$_{full}$ was purified using Hiload 26/600 Superdex 200 size-exclusion chromatography (Cytiva, Cat#28989336) equilibrated with 20 mM Tris-HCl (pH 7.5), 150 mM NaCl, and 5 mM dithiothreitol (DTT).

*Arabidopsis thaliana* MET1$_{MTase}$ (residues 621–1534) was subcloned into the pGEX-6P-1 plasmid (Cytiva, Cat#28954648). The protein fusing glutathione S-transferase (GST) was expressed in *Escherichia coli* (*E.coli*) Rosseta2 (DE3) (Novagen, Cat# 70954) in Luria–Bertani medium (LB). When the optical density at 660 nm (O.D.$_{660}$) of the cells reached 0.7, 0.2 mM isopropyl β-d-thiogalactoside (IPTG) was added to the medium and incubated at 15 °C for 15 h. The cells were suspended in lysis buffer (40 mM Tris-HCl (pH8.0) containing 300 mM NaCl, 30 μM Zn-acetate, 0.5 mM TCEP, 10% (w/v) glycerol, and protease inhibitor cocktail) and sonicated. After centrifugation, GST-tagged MET1$_{MTase}$ was purified with Glutathione Sepharose 4B (GS4B; GE Healthcare, Cat# 17075605), and the unbound proteins were washed with lysis buffer plus 1 M NaCl and lysis buffer. The protein was eluted using elution buffer (50 mM Tris-HCl (pH 8.0) containing 20 mM reduced glutathione (GSH), 300 mM NaCl, 1 mM DTT, and 10% (w/v) glycerol). The GST tag was cleaved by HRV3C protease at 4 °C for more than 12 h. MET1$_{MTase}$ was purified using HiTrap Heparin HP with a linear gradient from 50 to 1000 mM NaCl in 20 mM Tris-HCl (pH 8.0) buffer containing 0.5 mM DTT and 10% (w/v) glycerol. Finally, the protein was purified with Hiload 26/600 Superdex 200 equilibrated with 20 mM Tris-HCl (pH 7.5), 250 mM NaCl, 1 mM DTT, and 10% (w/v) glycerol. mutants of MET1$_{MTase}$ were generated by site-directed mutagenesis and purified in the same manner as the wild-type proteins.

cDNA of human MET1 encoding residues 57–332 (MET1$_{RFTS1}$), 339–616 (MET1$_{RFTS2}$), and 57–616 (MET1$_{RFTSI-2}$) were subcloned into the pGEX-6P-1 plasmid. The proteins were expressed in Rosetta 2 (DE3) cells. 0.2 mM IPTG was added to the cell when the O.D.$_{660}$ reached 0.7, and the cells were further cultured at 15 °C for 15 h. The cells were suspended in lysis buffer (40 mM Tris-HCl (pH8.0) containing 300 mM NaCl, 30 μM Zn-acetate, 0.5 mM TCEP, and 10% (w/v) glycerol) and sonicated. The supernatant was loaded onto GS4B. After the GST tag was removed using HRV3C protease, the sample was loaded onto HiTrap Q HP (Cytiva, Cat#17115301). Finally, the protein was purified with Hiload 26/600 Superdex 75 (Cytiva, Cat#28989334) equilibrated with 10 mM Tris-HCl (pH 7.5), 150 mM NaCl, 0.5 mM TCEP and 10 μM Zn-acetate.

Full-length *Arabidopsis thaliana* VIM1 (UniProt: Q8VYZ0) was subcloned into pGEX6P-1 and expressed as a GST-fusion protein. The proteins were expressed in Rosetta 2 (DE3) cells. 0.2 mM IPTG was added to the cells when O.D.$_{660}$ reached 0.7 and the cells were further cultured at 15 °C for 15 h. The cells were suspended in lysis buffer (40 mM Tris-HCl (pH8.0), 300 mM NaCl, 30 μM Zn-acetate, 0.5 mM TCEP, 10% (w/v) glycerol) and sonicated. The supernatant was loaded onto GS4B. The GST tag was cleaved by the HRV3C protease at 4 °C for more than 12 h. VIM1 was purified using HiTrap Heparin HP with a

 

linear gradient from 50 to 1000 mM NaCl in 20 mM Tris-HCl (pH 8.0) buffer containing 0.5 mM DTT and 10% (w/v) glycerol. Finally, the protein was purified using a Hiload 26/600 Superdex 200 equilibrated with 10 mM Tris-HCl (pH 7.5), 250 mM NaCl, 1 mM DTT, 10 μM Zn acetate, and 10% (w/v) glycerol.

Full-length *Arabidopsis thaliana* UBC11 (UniProt: P35134) was subcloned into pGEX6P-1 and expressed as GST-fusion protein. The proteins were expressed in Rosetta 2 (DE3) cells. 0.2 mM IPTG was added to the cells when $O.D._{660}$ reached 0.7 and the cells were further cultured at 15 °C for 15 hr. The cells were suspended with a lysis buffer (40 mM Tris-HCl (pH8.0), 300 mM NaCl, 2 mM EDTA, 0.5 mM TCEP, 5% (W/V) glycerol) and sonicated. The supernatant was loaded onto GS4B. HRV3C protease was added to the elution fraction to remove the GST tag, and the sample was loaded onto Hiload 26/600 Superdex 75 equilibrated with 10 mM Tris-HCl (pH 7.5), 150 mM NaCl, 0.5 mM TCEP.

Recombinant purified mouse UBA1 (E1, Uniprot: Q02053), human ubiquitin (Uniprot: P0CG48), and human histone H3.1 tail (residues 1–36 with an additional tryptophan residue at position 37: $H3_{1-37W}$, Uniprot: P68431) were prepared as described previously, and −80 °C stock samples were used in this study[77–79]. Ubiquitinated H3 analogs for the EMSA and in vitro DNA methylation were prepared as previous report, and −80 °C stock samples were used in this study[50]. Briefly, the target lysine residues in the $H3_{1-37W}$ were substituted with cysteine residues, and the G76C mutant ubiquitin was conjugated to the sulf-hydryl group of the cysteine via a disulfide bond.

### In vitro DNA methylation assay

The 42 base pairs of the DNA duplex (upper: 5′-GGACATCXGTGA-GATCGGAGGCXGCCTGCTGCAATCXGGTAG, X = unmethylated C or 5mC, 0-10 μM) were methylated with 100 nM MET1 and DNMT1 (wild type or mutants) by the addition of the 5 μM $H3_{1-37W}$ analogs ($H3_{WT}$, H3C14ub, H3C18ub, H3C23ub, H3C14ub/18ub, H3C18ub/C23ub, H3C14ub/C23ub, H3C14ub/C18ub/C23ub, and ubiquitin) including 20 μM S-adenosyl-L-methionine (SAM) in reaction buffer (20 mM Tris-HCl (pH 8.0), 50 mM NaCl, 1 mM EDTA, 3 mM $MgCl_2$, 0.1 mg/mL BSA, 20% glycerol) at 30 °C for 1 h. The methylation reaction and conversion of SAH to ADP were terminated by the addition of 5 × MTase-Glo™ reagent from the methyltransferase assay kit, MTase-Glo (Promega, Cat#V7602), at a 1:4 ratio for the total reaction volume. After 30 min of incubation at room temperature, ADP detection was carried out using solid white flat-bottom 96-well plates (Costar, Cat#3917). The MTase-Glo™ Detection Solution was added to the reaction in a 1:1 ratio to a reaction volume of 40 μL and incubated for 30 min at room temperature. Luminescence derived from the reaction product, SAH, was monitored using a GloMax® Navigator Microplate Luminometer (Promega, Cat#GM2000). Statistical significance was evaluated by a two-tailed Student's *t* tests. *P*-value < 0.05 was statistically significant.

### Thermal stability assay

Changes in the denaturation temperature of apo-$MET1_{full}$, $MET1_{MTase}$ and apo-$DNMT1_{full}$ were evaluated using SYPRO® Orange (Thermo Fisher Scientific, Cat#S6651). The assay was performed in 20 μL of 0.2 mg/ml protein in a buffer (20 mM Tris-HCl (pH7.5), 50 mM NaCl). The sample was heated from 25 °C to 90 °C in increments of 0.2 °C every 10 s using a CFX Connect™ Real-Time System (Bio-Rad, Cat#1855201) and a 96-well PCR plate (Bio-Rad, Cat#HSP9655). The measured fluorescence data were normalized to $(F(T) - F_{min})/(F_{max} - F_{min})$, where $F(T)$, $F_{max}$, $F_{min}$ represent each fluorescence intensity at a particular temperature, the maximum fluorescence intensity, and the minimum fluorescence intensity, respectively[80]. Three independent experiments were performed using the thermal stability assay.

### In vitro ubiquitination assay

Protein expression in *E. coli* and purification of mouse UBA1 (E1), *Arabidopsis thaliana* UBC11 (E2), *Arabidopsis thaliana* VIM1 (E3),

C-terminal FLAG tagged-$H3_{1-37W}$ and ubiquitin were performed according to previous reports[44]. The ubiquitination reaction mixtures contained 100 μM ubiquitin, 200 nM E1, 2 μM E2, 1.5 μM E3, 5 mM ATP, and 50 μM C-terminal FLAG tagged-$H3_{1-37W}$ in ubiquitination reaction buffer (50 mM Tris-HCl (pH 8.0), 50 mM NaCl, 5 mM $MgCl_2$, 0.1% Triton X-100, 2 mM DTT). The mixture was incubated at 30 °C for 6, 12, and 24 h and the reaction was stopped by adding 3 × SDS loading buffer. The reaction was analyzed by SDS-PAGE, followed by Western blotting using a 1/5000 diluted anti-FLAG antibody (Cell Signaling Technology, Cat#2368).

### Electrophoresis mobility shift assay

Samples (10 μL) were incubated for 30 min at 4 °C in a binding buffer (20 mM Tris-HCl (pH 7.5), 150 mM NaCl, 0.05% NP-40, 10% (w/v) glycerol), and electrophoresis was performed using 0.5 × Tris-borate buffer (25 mM Tris containing 12.5 mM boric acid (pH 8.8)) at a constant current of 8 mA for 120 min in a cold room on a 5-20% polyacrylamide gel (Wako, SuperSep™, Cat#194-15021). Ubiquitinated $H3_{1-37W}$ analogs (H3C18ub/C23ub) at 1-, 2-, and 3-fold molar excess were added to the sample solution containing 1 μM MET1 RFTS1, RFTS2, or the RFTS1–2 domains. Proteins were detected and analyzed by staining with Oriole™ (Bio-Rad, Cat#1610496) and ChemiDoc XRS system (Bio-Rad, Cat#1708265J1PC), respectively.

### Preparation of samples for cryo-EM single particle analysis

$MET1_{MTase}$ was mixed with 12 bp of hemimethylated DNA (upper: 5′-ACTTAMGGAAGG, lower: 5′-CCTTCFGTAAGT, M = 5-methylcytosine, F = 5-fluorocytosine) in the conjugation buffer (50 mM Tris-HCl (pH 7.5), 20% Glycerol, 5 mM DTT, 50 mM NaCl) for preparation of the complex. The reaction was initiated by the addition of 500 or 1000 μM SAM at 30 °C for 15 h. The yield was purified using Superdex® 200 Increase 10/300 GL (Cytiva, Cat#28990944) equilibrated with cryo-EM buffer (20 mM Tris-HCl (pH 7.5), 250 mM NaCl, 5 mM DTT) and concentrated to 0.47 mg/mL or 9.5 mg/mL. A carbon grid (Quantifoil Cu R1.2/1.3, 300 mesh, Cat# M2955C-1) was glow-discharged for 2 min in the HARD mode setting of PIB-10 (Vacuum Device, Inc.). 3 μL of the sample was applied to the grid and rapidly frozen in liquid ethane cooled with liquid nitrogen using a Vitrobot Mark IV (Thermo Fisher Scientific). To improve particle orientation, a sample solution containing a 9.5 mg/mL complex and 4 mM CHAPSO was added to the grid. 3 μL of the 0.7 mg/mL $MET1_{full}$ solution without CHAPSO was loaded onto the grid. The parameters for plunge-freezing were set as follows: blotting time, 3 s; waiting time, 3 s; blotting force, −5; humidity, 100%; and temperature, 4 °C. Data were collected using a 300 kV Titan Krios G4 (Thermo Fisher Scientific) in RIKEN, Yokohama, Japan, equipped with a K3-summit camera (Gatan) with a BioQuantum energy filter (slit width 15 eV). A total of 6823 and 15,038 movies of the $MET1_{MTase}$:hemimethylated DNA complex in the presence and absence of CHAPSO, respectively, were collected in counted mode (nonCDS) for 48 frames and an exposure time of 2.2 s with a total dose of 55.2 and 60.7 e⁻/Å², respectively. A total of 7479 movies of apo-$MET1_{full}$ were collected in nonCDS for 48 frames with a total dose of 54.1 e⁻/Å² and an exposure time of 2.6 s. The magnification of the micrographs was ×105,000, and the pixel size was 0.83 Å/px. Data were automatically acquired using the image shift method of EPU software (Version 3.2.0.4776REL and 3.7, Thermo Fisher Scientific) with a defocus range of −0.8 to −1.6 μm.

### Data processing

All data were processed using cryoSPARC v4.2.1 and v4.4.0[81]. Motion correction was performed with Patch Motion Correction and defocus values were estimated using the contrast transfer function (CTF) via the Patch CTF Estimation.

For $MET1_{MTase}$:hemimethylated DNA complex, a total of 6823 (CHAPSO-) and 15,038 (CHAPSO + ) micrographs were analyzed. Initial

data collection without CHAPSO exhibited a preferential orientation problem, and thus additional data were collected in the presence of CHAPSO to improve particle distribution. The 5,516,893 (CHAPSO-) and 10,333,570 (CHAPSO + ) particles were automatically picked by Blob Picker and extracted from micrographs with a box size of 256 px, which were reduced to 64 px (3.31 Å/px). After selecting the suitable particle classes from 2D classification, the two datasets were merged, total 2,304,720 particles. Four ab initio models were generated using Ab-initio Reconstruction and 2,304,720 particles were subsequently classified using two rounds of Heterogeneous Refinement. At this stage, 864,086 particles were selected for MET1$_{MTase}$:hemimethylated DNA complex and re-extracted with the 256 px box size and down-sampled to 230 px (0.92 Å/px). Finally, 2.74 Å resolution of the cryo-EM map was obtained using Non-uniform Refinement based on the gold-standard Fourier shell correlation (FSC = 0.143). The details of data processing are shown in Supplementary Fig. 3a. The final model was sharpened using a manually set B-factor of –100 using the sharpening tool of cryoSPARC.

For apo-MET1$_{full}$, a total of 7479 micrographs were analyzed, and 5,898,427 particles were automatically picked using Blob Picker. These particles were extracted from micrographs with a box size of 256 px, which were reduced to 64 px (3.31 Å/px) and subjected to 2D classifications. 1,009,984 particles were selected, and three ab initio models were generated from these particles. Four rounds of heterogeneous refinement were carried out, and good particles (410,338 particles) were re-extracted with the 256 px box size and down-sampled to 200 px (1.06 Å/px). Non-uniform Refinement of 410,338 particles yielded a cryo-EM map of 3.24 Å. To further clarify the model, 410,338 particles were subjected to 2D Classification. Then, three ab initio models were generated from the good particles (114,780 particles) selected from 2D Classification and Heterogeneous Refinement was performed on all particles (5,898,427 particles) using these models as references. After three rounds of Heterogeneous Refinement, the selected particles were extracted with a box size of 256 px, which were reduced to 128 px (1.66 Å/px). Three further rounds of Heterogeneous Refinement were performed, and 307,752 particles were selected for apo-MET1$_{full}$ and re-extracted with the 256 px box size and down-sampled to 200 px (1.06 Å/px). Finally, 3.17 Å resolution of the cryo-EM map was obtained using Non-uniform Refinement based on the criterion of FSC = 0.143. The details of data processing are shown in Supplementary Fig. 8a. The final model was sharpened using a manually set B-factor of –100.

## Model refinement

Structural model of the MET1$_{MTase}$:hemimethylated DNA complex was generated using AlphaFold2 (UniProt; P34881) and structural model of the apo-MET1$_{full}$ was generated using MET1$_{MTase}$:hemimethylated DNA complex. The model and electron microscopy maps were roughly fitted using UCSF ChimeraX version 1.7[82]. The model was constructed with Coot version 0.9.8.92[83] and further refined with Real-Space Refine in PHENIX version 1.21.2-5419 and version 1.20.1-4487[84]. Supplementary Table 1 presents data processing and refinement statistics. The RMSD is calculated using Matchmaker in UCSF ChimeraX.

## Sequence alignment

Multiple sequence alignments were performed using software MAFFT version 7 (https://mafft.cbrc.jp/alignment/server/index.html). *Arabidopsis thaliana* MET1 (UniProt; P34881), *Homo sapiens* DNMT1 (UniProt; P26358), *Mus musculus* DNMT1 (UniProt; P13864), *Xenopus tropicalis* DNMT1 (UniProt; F6QE78), *Danio rerio* DNMT1 (UniProt; A0A8M6YUX4), *Arabidopsis thaliana* MET2 (UniProt; O23273), *Oryza sativa subsp. Japonica* MET1A (UniProt; Q7Y1I7), *Pinus sylvestris* MET1-1 (UniProt; A0A2S1P6T3), Solanum lycopersicum MET1 (UniProt; A0A3Q7IUT4), *Vaccinium corymbosum* MET1 (UniProt; A0A1Z2RWY3) and *Zea mays* ZMET1 (UniProt; O65343) were used as input sequences.

## Reporting summary

Further information on research design is available in the Nature Portfolio Reporting Summary linked to this article.

## Data availability

Data supporting this study are available from the corresponding author upon reasonable request. The cryo-EM density maps have been deposited in the Electron Microscopy Data Bank (EMDB, www.ebi.ac.uk/pdbe/emdb/): EMD-63650 corresponds to the MET1$_{MTase}$:hemimethylated DNA complex, and EMD-63652 corresponds to apo-MET1$_{full}$. The corresponding atomic coordinates have been deposited in the Protein Data Bank (PDB, www.rcsb.org) under accession codes 9M5U and 9M5X, respectively. All data needed to evaluate the conclusions in the paper are presented in the paper and/ or Supplementary Materials. PDB 4WXX, 7XI9, 5WVO were used for this study. Source data are provided in this paper. Source data are provided with this paper.

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

## Acknowledgements

We thank Drs. Tetsuji Kakutani, Akihisa Osakabe, and Jafar Sharif for critical discussions on this research. The cryo-EM experiments were performed at the cryo-EM facility of the RIKEN Centre for Biosystems Dynamics in Yokohama, Japan, with special thanks to the members of the RIKEN BDR, particularly to Drs. Haruhiko Ehara and Yuko Murayama. The authors acknowledge the use of SciGen, a customized language model based on ChatGPT, for English editing and writing assistance during the preparation of this manuscript. No original scientific content or sentences were generated by the AI. This study was supported by MEXT/JSPS KAKENHI under Grant numbers 19H05741 and 24K01967, 24K21950, and 25H01301 to K.A. and 19H05285, 21H00272, and 24K02001 to A.N. and a grant for 2021–2023 and 2024–2026 Strategic Research Promotion (No. SK201904) from Yokohama City University to K.A.. A.K. was supported by Sasakawa Scientific Research Grant and JST SPRING, Japan (JPMJSP2179).

## Author contributions

K.A. supervised the study, wrote the manuscript, and prepared figures. T.TK. cloned the MET1 cDNA. A.K., A.N., Y.C., M.N., and K.A. designed and prepared expression vectors. A.K., A.N., Y.C., and K.A. performed protein purification. A.K. performed in vitro DNA methylation, ubiquitination, thermal stability assay, and EMSA assays. A.K. and K.A. performed cryo-EM experiments and analyses.

## Competing interests

The authors declare no competing interests.
