## [Transparent Peer Review file · Nature Communications]

Cryo-EM Reveals Evolutionarily Conserved and Distinct Structural Features of Plant CG Maintenance Methyltransferase MET1

Corresponding Author: Professor Kyohei Arita

Version 0:

Reviewer comments:

Reviewer #1

(Remarks to the Author)

In this work, Kikuchi and colleagues characterize the mechanism by which the plant DNA methyltransferase MET1 methylates hemimethylated DNA. The authors compare cryo-EM structures of MET1 in the apo-state and bound to DNA and show that the autoinhibited apo-state of MET1 can be alleviated upon binding of ubiquitinated histone H3. There are some similarities in the enzymatic mechanisms of human and plant methyltransferases but also important differences that have not been proposed or reported previously and are of substantial interest to the chromatin/epigenetics scientific community. Overall, the study is novel, exciting and very well executed. This manuscript contains excellent quality data, conclusions are convincing and justified, and I enthusiastically support publication.

A minor comment: please avoid using bold font and frames in Figures. best regards, Tatiana K.

Reviewer #2

(Remarks to the Author)

The manuscript from Kikuchi and colleagues reports the characterisation and structure of *A. thaliana* maintenance CG methyltransferase. The authors conclude that the active conformation of MET1 is like DNMT1, but that the inhibitory mechanism is different, showing the connecting loop to occupy the active site in the apo structure, instead of the RFTS domain.

Overall, this is a well written and presented piece of work that reveals a fundamental mechanism of how plants maintain CG methylation patterns. We thank the authors for including wwPDB validation reports, maps and models. This greatly aids the critical assessment of the data. The analysis and biochemistry for the most part is performed well. There are however some oddities in the way the EM data has been handled and written about in the experimental details in the methods and supplementary figures. We suggest that the below major comments are addressed prior to publication:

Major points

1. Particle orientation was clearly an issue due to the use of CHAPSO for second dataset. From manual inspection, there is some streaking in the density of the DNA-Mtase complex suggesting a degree of anisotropy. While this is probably not bad enough to obscure the results, this should be commented on and quantified. As a matter of good practice this should be reported for all cryo-EM structures, but especially important here. Please include 3D-FSC or equivalent in supp fig 3 and 7 and report quantitative measure of anisotropy such as SCF and cFAR measurements
2. Both maps have larger voxel sizes that do not match the pixel size stated for the detector. This is highly unusual. Was the microscope pixel size recalibrated? The boxsize seems to scale with the pixel size discrepancies suggesting that there will not be any appreciable effect on the overall scaling of the map and then subsequent model dimensions. (i.e. should be 256 box with 0.83 Å pixel). However, the correct way to show this data is fully unbinned or, if necessary, binned in a factorial

manner. Unusual binning such as this may produce artifacts in FSC values and even in the map that can lead to incorrect model building. Subtle inconsistencies in pixel size can lead to large artifacts in CTF estimation and all downstream processes, this is amplified when non-integer binning factors especially at higher spatial frequencies (i.e such as x1.277 and x1.111 utilised here).

These data should be re-extracted to native pixel size and reprocessed, these are then the maps that should be deposited reported.

3. VIM E3 ligase activity needs to be more robustly characterised in order to assert that this is a H3 N-terminal tail specific ubiquitin modifier. The assay was performed at high concentrations, very long time points, without controls and only on a short H3 tail peptide. The assay adds little to the manuscript at present and could be removed without altering the main conclusions. Performing ub reactions on just tails or individual histones must be treated with caution, it is common to get off-target ub when not in proper nucleosomal context as has been reported for many E3 ligases. Furthermore, in our experience E3 independent ubiquitylation can occur on lysine rich substrates in the conditions reported here. Controls such as K-R mutation at proposed ub sites, within a nucleosome, showing E3 ligases dependency at multiple timepoints would be required.

4. The methodology described for cryo-EM grid preparation/analysis needs expansion and re-consideration:

- Line 528, 530 what does “standard mode” mean for K3 operation? K3 collect data in Counting, super resolution and linear linear modes. With or without CDS applied?
- Line 542 What is “state binning state”? This is not a normal nomenclature for binning and particular important to clarify given the discrepancy in pixel sizes (see above)
- 522-528 make explicit that data was collected from two sperate grids.
- Further description of full-length grid conditions, were these with or without chapso, was the same volume applied to grid?
- Line 554 processing of full-length protein unclear. What is meant by “select particles again”? Would imply particle were repicked, but more likely that heterogenous refinement was used on full dataset of picked particles using the ab intio model. Also unclear from supplemental figure 7
- Supp fig 3 and data processing unclear how many rounds of heterogenous refinements, number of starting models/classes were used were performed. Would recommend showing all rounds and the maps that were chosen/removed via these steps and percentage of particles/no. of particles present. Also show the maps that were used during final non uniform refinement. This is important to see the manual assertion of discarded data.
- Supp fig 7 processing pipeline is difficult to follow. Multiple rounds of classification performed, before returning to the original picked 5.9 million picked particles data using only the ab intio model as a starting model? Big jump then from the 5.9million particle down to 307K that are in the final model. This is the part of processing that needs to be expanded on not how the initial model was generated. Recommend changing this figure to reflect the processing in a manner similar to supp fig S3. As for supp fig S3 show all maps from rounds of heterogenous refinements, with which classes were selected and their particle number or % distribution between classes.
- Report the B-factors applied to the maps, presumably these were sharpened?

Minor points

1. The suggested Connecting linker as autoinhibitory mechanism is interesting, biochemical assays using mutations or alterations to show this effect in Mtase assay would strengthen this hypothesis. This would be beneficial given the poorer density for this region in the structure.
2. The H3 ubiquitylated peptide stimulates activity, but it is a much smaller effect than the difference between MET1Mtase and MET1full (Fig 1), indeed the standard deviation in Supplemental Fig 1 is very large. The weight of evidence as split across Fig. 6D and Supp Fig S1D does seem to agree, but the authors should acknowledge this further. One possible hypothesis is that only H3 tail peptides were used rather than the more appropriate nucleosome substrates, which multiple studies have shown makes large affects for other chromatin binding proteins. Furthermore, binding to MET1 may not be saturated with the H3ub peptides given the multiple binding sites and lack of saturation shown for the RFTS domains alone. These possibilities and other hypothesis should be discussed in the manuscript.
3. Please include further map and model validation statistics in Table 1 and/or methods section. These should be included as standard for both casual and expert readers to assess the structural data. These should include
 - details about the reconstruction boxsize, voxel size, b-factor applied and how this was calculated. Include here also SCF, cFAR anisotropy measurements (see above)
 - Model vs data measurements such as CCmask and FSC model vs map
 - Model statistics such as CaBLAM outliers, ramaZ score, EMringer and molprobit score
4. Please include further details in the methods or Figure legends on EMSA assay (i.e Fig 6e), what concentrations/amounts were used?
5. What histone peptide sequence is used in this paper, is it H3.1 from human/xenopus or H3 sequence form Arabidopsis? Histone is used throughout when it is presumably the H3 N-terminal tail region? This should be clarified, for example line 456.
6. Supplemental Figure S1D and Figure 6D are the same assay and complementary to the authors conclusions, it is unclear

why they are separated? We recommend showing all of this data together, for simplicity allowing comparison between all 9 constructs tested perhaps as a bar chart reporting the activity at a single DNA concentration for all H3-ub peptides used in addition to the curves shown. Indeed, from comparing this data it is unclear why 14/18/23 ub does not stimulate further but 14/18/27 does? In the same assay, Ubiquitin alone and H3 tail alone would be nice controls.

Reviewer #3

(Remarks to the Author)

Reviewer #4

(Remarks to the Author)

In "Cryo-EM reveals evolutionarily conserved and Distinct structural features of plant CG maintenance methyltransferase MET1" Amika Kikuchi et al., analyzed the molecular structure of plant DNA methyltransferase MET1 in both Apo and DNA-bound forms. In comparison with DNMT1, they found that the autoinhibitory state of MET1 differs from that of DNMT1, although its overall structure and activation mechanism are similar to those of mammalian DNA methyltransferase. Additionally, the authors also demonstrated that MET1 can be activated by binding to ubiquitinated H3, a mechanism that also resembles that of mammalian DNMT1. Because MET1 is critical for maintenance of CG methylation in plants, its specific regulation mechanism should be studied in depth. In this manuscript, the data quality and structural analysis are solid and meticulous. It involves a certain amount of work and the experiments were well executed. However, the biological insight and novelty from this work is not as impressive. The majority of the result focuses on describing the structural and mechanistic similarities between MET1 and DNMT1, while lacking in-depth investigation on the unique characteristics of plant DNA demethylases. The results of this work will be useful in understanding how plant DNA methyltransferase MET1 work under active or inactive state, and the experimental structures should be a valuable resource for future studies. I cannot state whether this results fit the level of the journal or not, it is preferring to leave to the Editor for evaluation. I do not have any experimentally based advice to offer, as the data are compelling and well presented.

Version 1:

Reviewer comments:

Reviewer #2

(Remarks to the Author)

The revised manuscript is improved and an important and overall very nice piece of work. We believe it is nearly suitable for publication, and we are grateful to the authors for addressing the majority of points. The methods, cryo-EM processing figures and supplemental table are greatly improved. We have one point more for our own interest and understanding and one scientific point that we believe needs addressing:

H3 ubiquitylation

We are appreciative of the new data using plant E2 and agree that the findings does add value. However, The author state:

"H3.1 N-terminal tail (aa: 1-37W, H31-37W) undergoes VIM1-dependent (sic) ubiquitination (Fig. 6c and Supplementary Fig. 9a)."

In the data presented this does not appear to be the case, as VIM 1 independent H3ub1 is observed (figure 6), a band is observed for H3ub1 in the absence of the E3. Indeed, this was what we were concerned about our original comment 3, which was not fully addressed in the revised manuscript nor rebuttal. Reaction times and enzyme concentrations are very high and we would be worried this would be far from the expected physiological situation. We do agree that VIM1 can promote further ubiquitylation of these tails but worry that this has not been proven and would only end up hurting further studies of its function. We agree that a full exploration of Vim1 activity is outside of the scope of this study and looking forward to seeing the authors upcoming work on this.

We think it is essential given the data presented the caveats and likely artefactual nature of the H3-ub1 observed is acknowledged in the text to not undermine future work on VIM1 E3 ligases activity, unless the authors believe H3ub1 is E3 independent?

Odd voxel sizes

We were of course aware that the data was Fourier cropped, we use binning as this was the nomenclature used in the previous manuscript in the somewhat confused methodology. The methods are now far better explained using the correct terminology and clearer as to the steps performed.

It is still somewhat odd to use an arbitrary Fourier cropping size, and not as suggested common in the cryo-EM field in our experience. More for our own interest it would be nice to know why such unusual number was used. Re-extracting to the original boxsize should lead to increased detail in the maps, not requiring Fourier cropping. Was this computational necessity, historical or did this lead to increased signal to noise at the artificial voxel size? We note the cFAR value for map 2 is quite low (only just over the 0.5 threshold for anisotropy) while SCF* is high, this is usually indicative of contaminant junk particles in the final model and further 3D classification may help.

Reviewer #3

(Remarks to the Author)

Reviewer #4

(Remarks to the Author)

In the revised version of "Cryo-EM Reveals Evolutionarily Conserved and Distinct Structural Features of Plant CG Maintenance Methyltransferase MET1", Amika Kikuchi et al., have further expanded and refined the manuscript content, including the plant conservation analysis, complemented in vitro ubiquitination assay and enhanced the methodological description, as well as added schematic representation of MET1, all of which have effectively improved the manuscript quality.

However, I still have a couple of minor questions:

1. In line 153, the authors listed 6 residues responsible for 5mC recognition. However, in the corresponding Fig.3b only W1438 was mutated to demonstrate the reduced DNA methylation activity. The main text does not provide an explanation for focusing solely on this residue, I just wonder the reason.
2. In line 191, the authors stated that "amino acid residues...are conserved between mammals and plants", therefore, in Fig. S7 the conserved residues should be clearly labeled out as was done in Fig S4.
3. In line 332, the authors claimed that "AlphaFold failed to predict the RFTS2 orientation of MET1". To better illustrate this discrepancy, I suggest comparing the Cryo-EM structure of MET1 with AlphaFold prediction to visually demonstrate the differences.
4. In Fig 6d, the authors demonstrated that mono-, di- and tri- ubiquitinated histone tails exhibit distinct DNA methylation rates. However, I am curious about the rationale for specifically selecting K14, K18, K23 and K27 for analysis. Given the statement in line 372 that "It is not yet known which histones are the targets for ubiquitination by VIM1". do these chosen residues reflect biologically meaningful ubiquitination sites in plants?

Version 2:

Reviewer comments:

Reviewer #2

(Remarks to the Author)

The authors have addressed all our points sufficiently and we support this manuscript for publication

Reviewer #3

(Remarks to the Author)

Thank you very much for taking the time to evaluate our manuscript. We sincerely appreciate the reviewers' constructive comments and suggestions. In response, we have carefully revised the manuscript accordingly. Please find below our detailed point-by-point responses to each of the reviewers' comments. Revised and newly added sentences are indicated in red in the manuscript.

Reviewer #1 (Remarks to the Author)

In this work, Kikuchi and colleagues characterize the mechanism by which the plant DNA methyltransferase MET1 methylates hemimethylated DNA. The authors compare cryo-EM structures of MET1 in the apo-state and bound to DNA and show that the autoinhibited apo-state of MET1 can be alleviated upon binding of ubiquitinated histone H3. There are some similarities in the enzymatic mechanisms of human and plant methyltransferases but also important differences that have not been proposed or reported previously and are of substantial interest to the chromatin/epigenetics scientific community. Overall, the study is novel, exciting and very well executed. This manuscript contains excellent quality data, conclusions are convincing and justified, and I enthusiastically support publication.

A minor comment: please avoid using bold font and frames in Figures. best regards, Tatiana K.

>Dear Dr. Tatiana.

We are grateful for your encouraging and supportive feedback throughout the review process. Following your thoughtful suggestions, we have revised the figures accordingly in the updated version of the manuscript.

Reviewer #2 (Remarks to the Author)

The manuscript from Kikuchi and colleagues reports the characterisation and structure of A. thaliana maintenance CG methyltransferase. The authors conclude that the active conformation of MET1 is like DNMT1, but that the inhibitory mechanism is different, showing the connecting loop to occupy the active site in the apo structure, instead of the RFTS domain.

Overall, this is a well written and presented piece of work that reveals a fundamental mechanism of how plants maintain CG methylation patterns. We thank the authors for including wwPDB validation reports, maps and models. This greatly aids the critical

assessment of the data. The analysis and biochemistry for the most part is performed well. There are however some oddities in the way the EM data has been handled and written about in the experimental details in the methods and supplementary figures. We suggest that the below major comments are addressed prior to publication:

Major points

1. Particle orientation was clearly an issue due to the use of CHAPSO for second dataset. From manual inspection, there is some streaking in the density of the DNA-Mtase complex suggesting a degree of anisotropy. While this is probably not bad enough to obscure the results, this should be commented on and quantified. As a matter of good practice this should be reported for all cryo-EM structures, but especially important here. Please include 3D-FSC or equivalent in supp fig 3 and 7 and report quantitative measure of anisotropy such as SCF and cFAR measurements

> We appreciate your suggestions. In response, we have included the 3D-FSC and cFAR analyses in the revised Supplementary Figures 3 and 8 and Supplementary Table1.

2. Both maps have larger voxel sizes that do not match the pixel size stated for the detector. This is highly unusual. Was the microscope pixel size recalibrated? The boxsize seems to scale with the pixel size discrepancies suggesting that there will not be any appreciable effect on the overall scaling of the map and then subsequent model dimensions. (i.e. should be 256 box with 0.83 Å pixel). However, the correct way to show this data is fully unbinned or, if necessary, binned in a factorial manner. Unusual binning such as this may produce artifacts in FSC values and even in the map that can lead to incorrect model building. Subtle inconsistencies in pixel size can lead to large artifacts in CTF estimation and all downstream processes, this is amplified when non-integer binning factors especially at higher spatial frequencies (i.e. such as x1.277 and x1.111 utilised here).

These data should be re-extracted to native pixel size and reprocessed, these are then the maps that should be deposited reported.

>Thank you for the suggestions. We would like to emphasize that the down-sampling in our study was performed using Fourier cropping, not binning. This method is a standard approach in cryo-EM data processing and does not involve interpolation or averaging across pixels, as occurs in conventional binning. Consequently, it does not produce the artifacts that

the reviewer is concerned about, such as those affecting FSC calculations, CTF estimation, or model building. However, the previous Supplementary Figure regarding workflow of cryo-EM data may have caused misunderstanding. We apologize for any confusion this may have caused. We have amended the final extract process as shown in 256 px, binned to 200 px in the method section and Supplementary Figures 3 and 8.

3. VIM E3 ligase activity needs to be more robustly characterised in order to assert that this is a H3 N-terminal tail specific ubiquitin modifier. The assay was performed at high concentrations, very long time points, without controls and only on a short H3 tail peptide. The assay adds little to the manuscript at present and could be removed without altering the main conclusions. Performing ub reactions on just tails or individual histones must be treated with caution, it is common to get off-target ub when not in proper nucleosomal context as has been reported for many E3 ligases. Furthermore, in our experience E3 independent ubiquitylation can occur on lysine rich substrates in the conditions reported here. Controls such as K-R mutation at proposed ub sites, within a nucleosome, showing E3 ligases dependency at multiple timepoints would be required.

> VIM proteins play a pivotal role in the maintenance of CG DNA methylation in plants, as knockout mutants exhibit severe hypomethylation phenotypes similar to those observed in MET1 knockout lines. However, to date, there have been no reports identifying specific substrates of VIM proteins. In the present study, we conducted in vitro ubiquitination assays, inspired by mechanistic analogies with mammalian DNA methylation maintenance. We found that VIM1 catalyzes multiple monoubiquitinations on the N-terminal tail of histone H3, resembling the activity of mammalian UHRF1. We believe this finding provides new insight into the mechanism of DNA methylation maintenance in plants, and therefore, we have retained this data in the revised manuscript.

In response to the reviewer's comment, we have performed in vitro ubiquitination assays using *Arabidopsis thaliana* UBC11, which has been identified as E2 enzyme for VIM proteins (Kraft et al., *Plant J.* 2008). We also examined the time courses of the reaction and E2 enzyme concentrations to minimize potential off-target ubiquitination of histone H3. The new data was presented in Figure 6C in the revised manuscript.

Lines 297-304

In vitro ubiquitination assays using mouse UBA1, *Arabidopsis thaliana* UBC11, and VIM1 demonstrated that histone H3.1 N-terminal tail (aa: 1-37W, H31-37W) undergoes VIM1-

dependent ubiquitination (Fig. 6c and Supplementary Fig. 9a). *In vitro* ubiquitination assay using a K/R ubiquitin mutant, in which all lysine residues were replaced with arginine residues to prevent poly-ubiquitination, showed a ubiquitination pattern similar to that of wild-type ubiquitin, indicating that VIM1 catalyzes multiple mono-ubiquitination of histone H3 tail (Fig. 6c)'

We also appreciate the suggestion regarding the use of nucleosomes in the ubiquitination assays. VIM1 is multidomain protein composed of a PHD finger, RING1 domain, SRA domain, and RING2 domain. The RING domains are responsible for its ubiquitin ligase activity. According to our unpublished data, the SRA domain preferentially recognizes hemimethylated DNA. The PHD finger, on the other hand, has been reported not to bind to the histones. Therefore, when designing nucleosome-based assays, we have to carefully consider the functional coordination among these domains—particularly how the SRA domain accesses hemimethylated sites. Should linker DNA or nucleosomal DNA be included? Which histones are targeted by VIM1-mediated ubiquitination? What role, if any, does the PHD finger play in this process? These questions remain open and require further investigation.

Given the complexity and scope of these questions, we believe that addressing them thoroughly falls beyond the scope of the present study. We plan to explore these important issues in future research.

We now address the potential ubiquitination of nucleosomes by VIM1 in the revised manuscript.

Lines 338-342

'The relatively modest enhancement of MET1 activity by the dual mono-ubiquitinated H3 may be attributed to an insufficient amount of H3ub2 to fully saturate MET1 binding, as well as the use of an H3 tail peptide rather than a more physiologically relevant nucleosome substrate.'

We apologize if this was unclear in the original revision, but we would like to point out that we have already discussed the possibility of VIM-mediated ubiquitination of other histones in lines 372–378 of the revised manuscript.

4. The methodology described for cryo-EM grid preparation/analysis needs expansion and re-consideration:

- Line 528, 530 what does “standard mode” mean for K3 operation? K3 collect data in Counting, super resolution and linear linear modes. With or without CDS applied?
- Line 542 What is “state binning state”? This is not a normal nomenclature for binning and particular important to clarify given the discrepancy in pixel sizes (see above)
- 522-528 make explicit that data was collected form two sperate grids.
- Further description of full-length grid conditions, were these with or without chapso, was the same volume applied to grid?
- Line 554 processing of full-length protein unclear. What is meant by “select particles again”? Would imply particle were repicked, but more likely that heterogenous refinement was used on full dataset of picked particles using the ab intio model. Also unclear from supplemental figure 7
- Supp fig 3 and data processing unclear how many rounds of heterogenous refinements, number of starting models/classes were used were performed. Would recommend showing all rounds and the maps that were chosen/removed via these steps and percentage of particles/no. of particles present. Also show the maps that were used during final non uniform refinement. This is important to see the manual assertion of discarded data.
- Supp fig 7 processing pipeline is difficult to follow. Multiple rounds of classification performed, before returning to the original picked 5.9 million picked particles data using only the ab intio model as a starting model? Big jump then from the 5.9million particle down to 307K that are in the final model. This is the part of processing that needs to be expanded on not how the initial model was generated. Recommend changing this figure to reflect the processing in a manner similar to supp fig S3. As for supp fig S3 show all maps from rounds of heterogenous refinements, with which classes were selected and their particle number or % distribution between classes.
- Report the B-factors applied to the maps, presumably these were sharpened?

>We appreciate your valuable feedback regarding the statistical presentation of the cryo-EM data. In response, we have updated Supplementary Table 1 to include the appropriate information and amended Supplementary Figure 3 and 8.

Minor points

1. The suggested Connecting linker as autoinhibitory mechanism is interesting, biochemical assays using mutations or alterations to show this effect in Mtase assay would strengthen this hypothesis. This would be beneficial given the poorer density for this region in the structure.

>We greatly appreciate your insightful comment.

Our structural data suggest that the autoinhibitory function mainly arises from a steric clash between the RFTS2 domain and DNA. The connecting linker contains a highly conserved NLNPxA motif across plants, and the side chains of the two asparagine residues form hydrogen bonds with the RFTS2 domain. However, most interactions between the connecting linker and the RFTS2 domain are van der Waals contacts, and we suspect that point mutations in the linker may not substantially affect the RFTS2 positioning. Moreover, altering the linker length could potentially disrupt overall protein folding rather than selectively modulate its autoinhibitory role. We are highly interested in experimentally validating these possibilities and will pursue biochemical assays, particularly after identifying relevant activating factors.

We have added the below text to the revised manuscript.

Lines 281-284

'Although the connecting linker sequence is poorly conserved among plants, except for the NLNPxA motif (Fig. 4d and Supplementary Fig. 4b), the linker length is conserved to within one residue. This observation suggests that the physical length of the linker, rather than its specific sequence, is essential for inhibiting DNA binding.'

2. The H3 ubiquitylated peptide stimulates activity, but it is a much smaller effect than the difference between MET1Mtase and MET1full (Fig 1), indeed the standard deviation in Supplemental Fig 1 is very large. The weight of evidence as split across Fig. 6D and Supp Fig S1D does seem to agree, but the authors should acknowledge this further. One possible hypothesis is that only H3 tail peptides were used rather than the more appropriate nucleosome substrates, which multiple studies have shown makes large affects for other chromatin binding proteins. Furthermore, binding to MET1 may not be saturated with the H3ub peptides given the multiple binding sites and lack of saturation shown for the RFTS domains alone. These possibilities and other hypothesis should be discussed in the manuscript.

> In response to your suggestions, we have amended the text to explicitly acknowledge the limitations of our experimental design, specifically the use of the histone H3 tail peptide rather than the more physiologically relevant nucleosome substrate, as well as the possibility that the amount of H3ub2 used was insufficient to achieve full saturation of MET1 binding. We have added the following text to the Discussion section of the revised manuscript:

Lines 338-342

'The relatively modest enhancement of MET1 activity by the dual mono-ubiquitinated H3 may be attributed to an insufficient amount of H3ub2 to fully saturate MET1 binding, as well as the use of an H3 tail peptide rather than a more physiologically relevant nucleosome substrate.'

We fully agree that the regulation of MET1 activity in cells is likely more complex than assumed. To address this point, we have added a potential MET1 binding partner that may modulate MET1 activity. We have added the following text to the revised manuscript.

Lines 348-350

'In addition, histone-lysine N-methyltransferase MEDEA (MEA) is also known to directly interact with N-terminal region of MET1 *in vitro*, suggesting a possibility of functional modulation of the RFTS domains.'

3. Please include further map and model validation statistics in Table 1 and/or methods section. These should be included as standard for both casual and expert readers to assess the structural data. These should include

- details about the reconstruction boxsize, voxel size, b-factor applied and how this was calculated. Include here also SCF, cFAR anisotropy measurements (see above)*
- Model vs data measurements such as CCmask and FSC model vs map*
- Model statistics such as CaBLAM outliers, ramaZ score, EMringer and molprobity score*

>Thank you very much for your helpful suggestions. We have incorporated the requested information into the revised Supplementary Table 1.

4. Please include further details in the methods or Figure legends on EMSA assay (i.e Fig 6e), what concentrations/amounts were used?

>Thank you for the helpful suggestion. We have added detailed information regarding the EMSA assay, including the concentrations and amounts used, to the Methods section.

Lines 533-535

'Ubiquitinated H3_{1-37W} analogs (H3C18ub/C23ub) at 1-, 2-, and 3-fold molar excess were added to the sample solution containing 1 μ M MET1 RFTS1, RFTS2, or the RFTS1-2

domains.'

5. *What histone peptide sequence is used in this paper, is it H3.1 from human/xenopus or H3 sequence from Arabidopsis? Histone is used throughout when it is presumably the H3 N-terminal tail region? This should be clarified, for example line 456.*

>Thank you for pointing this out. In this study, we used the N-terminal tail of human histone H3.1 because it is fully conserved between human and *Arabidopsis thaliana*. We have clarified this information in the text, as well as throughout the manuscript where appropriate.

6. *Supplemental Figure S1D and Figure 6D are the same assay and complementary to the authors conclusions, it is unclear why they are separated? We recommend showing all of this data together, for simplicity allowing comparison between all 9 constructs tested perhaps as a bar chart reporting the activity at a single DNA concentration for all H3-ub peptides used in addition to the curves shown. Indeed, from comparing this data it is unclear why 14/18/23 ub does not stimulate further but 14/18/27 does? In the same assay, Ubiquitin alone and H3 tail alone would be nice controls.*

> Thank you for the comments. In response, we have performed in vitro DNA methylation assay of MET1 in the presence of H3_{1-37W} or ubiquitin, indicating that neither protein was able to activate MET1. In addition, we have prepared the bar graph presentation using the endpoint data at end points (10 μ M DNA). The revised figures have been added to Figure 6d and Supplementary Figure 1e in the revised manuscript.

Reviewer #3 (Remarks to the Author)

> Thank you for your evaluation of our manuscript. We appreciate your efforts as part of the Nature Communications initiative to support peer review training. We hope that our work contributed positively to your experience in the review process.

Reviewer #4 (Remarks to the Author):

This study reported cryogenic electron microscopy (cryo-EM) structures of MET1 in both apo and DNA-bound states. The activation and autoinhibitory mechanism of MET1 are based on the structural analysis.

1. In the "Enzymatic activity and substrate specificity of Arabidopsis thaliana MET1" section, if possible, it is recommended to compare the activity of MET1 with other plant (such as tomatoes) MET1 to show the unique or common characteristics of MET1.

> Thank you for the suggestion. In response, we performed a multiple sequence alignment of plant MET1 proteins. As shown in Supplementary Figure 4a in the revised manuscript, the amino acid residues involved in the recognition of 5mC within the TRD domain, as well as those contacting the target cytosine, are highly conserved. This strongly supports the functional conservation of MET1 activity across plant species. We have included a corresponding statement in the revised manuscript

Lines 162-164

'The amino acid residues responsible for recognizing 5mCG and 5fC in Arabidopsis thaliana MET1 are highly conserved among MET1 orthologs in other plant species, suggesting that plant MET1 proteins specifically recognize hemimethylated CG sites (Supplementary Fig. 4a).'

2. In the "Role of RFTS domains in MET1 activation" section, more discussion on the interaction between the RFTS domain and potential activators-ubiquitinated histone H3(Fig.6) can be added, which is good for understanding the activation mechanism of MET1. It is suggested this interaction be presented graphically to help readers better understand the mechanism.

>In response to the reviewer's suggestion, we have added a schematic representation illustrating the potential mechanism of MET1 activation mediated by ubiquitinated histone H3 or other unknown factors. This has been included as a new Figure 7 in the revised manuscript to help readers better understand the proposed model.

3. In the discussion of " H3ub2 promoted the DNA methylation activity of MET1 by only 2-fold", please provide more clarity. Is there a p-value or other statistical measure indicating the significance of the difference between the effects of H3ub2 on MET1 versus MET1MTase? Were the conditions for MET1 and DNMT1 assays matched in terms of enzyme concentration, reaction time, and substrate used?

>Thank you for the comment. To clarify the enhancement of DNA methylation activity by H3ub2, we have added a bar graph (Figure 6d in the revised manuscript) presenting the endpoint methylation levels at 10 μ M DNA substrate, including statistical analysis using Student's t-test. The observed ~2-fold increase in MET1 activity upon H3ub2 stimulation was statistically significant ($p < 0.05$). Regarding the assay conditions, the MET1 and DNMT1 assays were performed under identical experimental conditions, including enzyme concentration, reaction time, and substrate. The detailed information has been described in the revised Methods section as follows:

Lines 489-494

'The 42 base pairs of the DNA duplex (upper: 5'-GGACATCXGTGAGATCGGAGGCXGCCTGCTGCAATCXGGTAG, X = unmethylated C or 5mC, 0-10 μ M) were methylated with 100 nM MET1 and DNMT1 (wild type or mutants) by the addition of the 5 μ M H31-37W analogs (H3WT, H3C14ub, H3C18ub, H3C23ub, H3C14ub/18ub, H3C18ub/C23ub, H3C14ub/C23ub, H3C14ub/C18ub/C23ub, H3C18ub/C23ub/C27ub, and ubiquitin) including 20 μ M S-adenosyl-L-methionine (SAM) in reaction buffer (20 mM Tris-HCl (pH 8.0), 50 mM NaCl, 1 mM EDTA, 3 mM MgCl₂, 0.1 mg/mL BSA, and 20% glycerol) at 30°C for 1 h.'

Thank you very much for the continued evaluation of our manuscript. We sincerely appreciate the reviewers' constructive comments and valuable suggestions.

Reviewer #2 (Remarks to the Author):

The revised manuscript is improved and an important and overall very nice piece of work. We believe it is nearly suitable for publication, and we are grateful to the authors for addressing the majority of points. The methods, cryo-EM processing figures and supplemental table are greatly improved. We have one point more for our own interest and understanding and one scientific point that we believe needs addressing:

H3 ubiquitylation

We are appreciative of the new data using plant E2 and agree that the findings does add value. However, The author state:

“H3.1 N-terminal tail (aa: 1-37W, H31-37W) undergoes VIM1-dependet (sic) ubiquitination (Fig. 6c and Supplementary Fig. 9a).”

In the data presented this does not appear to be the case, as VIM 1 independent H3ub1 is observed (figure 6), a band is observed for H3ub1 in the absence of the E3. Indeed, this was what we were concerned about our original comment 3, which was not fully addressed in the revised manuscript nor rebuttal. Reaction times and enzyme concentrations are very high and we would be worried this would be far from the expected physiological situation. We do agree tha VIM1 can promote further ubiquitylation of these tails but worry that this has not been proven and would only end up hurting further studies of its function. We agree that a full exploration of Vim1 activity is outside of the scope of this study and looking forward to seeing the authors upcoming work on this.

We think it is essential given the data presented the caveats and likely artefactual nature of the H3-ub1 observed is acknowledged in the text to not undermine future work on VIM1 E3 ligases activity, unless the authors believe H3ub1 is E3 independent?

>Thank you very much for the comment. We agree that our previous statement may have overstated the implications. To avoid overstatement, we have rephrased the sentence to reflect our findings. We have carefully checked the overstatements in the previous manuscript, and modified them as follows:

Line 299

‘In vitro ubiquitination assays using mouse UBA1, Arabidopsis thaliana UBC11, and VIM1 demonstrated that that VIM1 promoted ubiquitination of the N-terminal tail of human histone H3.1 (aa: 1–37W, H3_{1–37W}), a region fully conserved between human and plants (Fig. 6c and Supplementary Fig. 9a). Initial ubiquitination appeared to be non-specifically catalyzed by the E2 enzyme, potentially

due to suboptimal experimental conditions, such as the use of an E2 concentration higher E2 than the physiological level.'

Line 310

'Based on these results, we hypothesized that ~~VIM proteins dependent~~ ubiquitinated histone H3 tail, catalyzed by VIM1 proteins, engages in the release of MET1 autoinhibition by binding to the RFTS domains.'

We have refined the experimental conditions by adjusting the E2 concentrations and reaction time. We found that at lower E2 concentrations, ubiquitination of the histone H3 tail was barely detectable. Although the E2 concentrations used in our experiments are indeed higher than physiological levels, they are required under our current *in vitro* conditions to enable efficient and reproducible ubiquitination of histone H3 by VIM1, allowing us to evaluate its enzymatic activity.

Odd voxel sizes

We were of course aware that the data was Fourier cropped, we use binning as this was the nomenclature used in the previous manuscript in the somewhat confused methodology. The methods are now far better explained using the correct terminology and clearer as to the steps performed.

It is still somewhat odd to use an arbitrary Fourier cropping size, and not as suggested common in the cryo-EM field in our experience. More for our own interest it would be nice to know why such unusual number was used. Re-extracting to the original boxsize should lead to increased detail in the maps, not requiring Fourier cropping.

Was this computational necessity, historical or did this lead to increased signal to noise at the artificial voxel size? We note the cFAR value for map 2 is quite low (only just over the 0.5 threshold for anisotropy) while SCF is high, this is usually indicative of contaminant junk particles in the final model and further 3D classification may help.*

> Thank you very much for the valuable comments and for raising the issue regarding voxel size selection and Fourier cropping.

Due to facility constraints (specifically, the fact that our PC cluster is shared among five other research groups), we were compelled to manage both computational resources and disk space with great care. Consequently, we reduced the dataset size by applying Fourier cropping, while still ensuring sufficient resolution for structural analysis. Given an initial box size of 256 pixels and a pixel size of 0.83 Å, the box size X (px) was calculated as follows, in which Y represents the target resolution:

$$X \text{ (px)} = (0.83 \text{ (Å)} \times 2) / Y \text{ (Å)} \times 256 \text{ (px)}$$

We down-sampled particles to the calculated box size and performed Non-uniform Refinement. When

the resulting FSC curve did not drop to zero before reaching the Nyquist limit, we reset the target resolution to a higher value and recalculated the box size.

For the MET1_{MTase}:hemimethylated DNA complex, the initial Non-uniform Refinement was performed with particles down-sampled from 256 px to 128 px (1.66 Å/px), and the FSC curve did not fall to zero before the Nyquist limit as shown in below figure.

Next, we set the target resolution of 3.0 Å ($X(\text{px}) = (0.83(\text{Å}) \times 2) / 3.0(\text{Å}) \times 256(\text{px}) = 141.65$). Particles were re-extracted with the 256 px box size and down-sampled to 150 px (1.42 Å/px) and Non-uniform Refinement was conducted.

Finally, since the FSC curve again did not drop to zero before the Nyquist limit, we utilized the high frequency information to reconstruct the map. We set the target resolution to 2.0 Å and down-sampled the data to 230 px (0.92 Å/px), which enabled us to check for duplicate particles. After Non-uniform Refinement, the FSC curve reached zero well before the Nyquist limit (Supplementary Figure 3).

For the refinement of apo-MET1_{full}, the initial Non-uniform Refinement was performed using particles down-sampled to 128 px (1.66 Å/px). Since the FSC curve did not drop to zero, we set the target resolution to 2.2 Å. Finally, Non-uniform Refinement was performed with particles down-sampled from 256 px to 200 px (1.06 Å/px), and FSC curve reached zero before the Nyquist limit (Supplementary Figure 8).

In both cases, we acknowledge—thanks to your insightful comment—that we may have overestimated the target resolution, which led to the unnecessary use of high-frequency information during refinement. We sincerely appreciate your valuable observation, which helped us recognize this issue.

In response to the second comment, we performed Heterogeneous Refinement and 3D classification to remove junk particles from the dataset. After particle re-extraction and Non-uniform Refinement, the cFAR value improved to 0.55. However, this process did not lead to a significant improvement of the map around the RFTS2 domain or other ambiguous regions. Therefore, we would like to use the previous map as the final version in this study. We sincerely appreciate your suggestions; the cFAR and SCF evaluation will be valuable for our future study.

Reviewer #3 (Remarks to the Author):

> Thank you for your continued evaluation and constructive feedback on our manuscript.

Reviewer #4 (Remarks to the Author):

In the revised version of “Cryo-EM Reveals Evolutionarily Conserved and Distinct Structural Features of Plant CG Maintenance Methyltransferase MET1 ”, Amika Kikuchi et al., have further expanded and refined the manuscript content, including the plant conservation analysis, complemented in vitro ubiquitination assay and enhanced the methodological description, as well as added schematic representation of MET1, all of which have effectively improved the manuscript quality. However, I still have a couple of minor questions:

1. In line 153, the authors listed 6 residues responsible for 5mC recognition. However, in the corresponding Fig.3b only W1438 was mutated to demonstrate the reduced DNA methylation activity. The main text does not provide an explanation for focusing solely on this residue, I just wonder the reason.

> Thank you very much for the comment. We selected W1438 based on a previous crystallographic

study of mammalian DNMT1, which showed that mutation of the corresponding residue, W1512, significantly reduces the DNA methylation activity of DNMT1 (Song et al., *Science* 2012). To clarify the rationale for selecting only W1438, we have added the following sentence in the revised manuscript.

Line 197

‘This is consistent with findings in mammalian DNMT1, where mutation of the corresponding residue, W1512 in mice, also abolished methylation activity⁶⁰.’

2. In line 191, the authors stated that “amino acid residues...are conserved between mammals and plants”, therefore, in Fig. S7 the conserved residues should be clearly labeled out as was done in Fig S4.

> Thank you for the comment. As suggested, we have added the label in Supplementary Figure 7.

In addition, since we compared the amino acid sequences between human DNMT1 and *Arabidopsis thaliana* MET1, we have amended the sentence on line 192 (line 191 in the previous version of the manuscript) as follows:

Line 192

*‘As the amino acid residues involved in DNA methylation are conserved between **human DNMT1 and Arabidopsis thaliana MET1** (Supplementary Fig. 7),’*

3. In line 332, the authors claimed that “AlphaFold failed to predict the RFTS2 orientation of MET1”. To better illustrate this discrepancy, I suggest comparing the Cryo-EM structure of MET1 with AlphaFold prediction to visually demonstrate the differences.

> Thank you for the valuable comment. In response, we have prepared a figure comparing the cryo-EM structure with the AlphaFold predicted structure. Please find the Supplementary Figure 10a in the revised version.

4. In Fig 6d, the authors demonstrated that mono-, di- and tri- ubiquitinated histone tails exhibit distinct DNA methylation rates. However, I am curious about the rationale for specifically selecting K14, K18, K23 and K27 for analysis. Given the statement in line 372 that “It is not yet known which histones are the targets for ubiquitination by VIMI”. do these chosen residues reflect biologically meaningful ubiquitination sites in plants?

> Thank you very much for your helpful comment.

The candidate residues for ubiquitination were selected based on known H3 ubiquitination sites by UHRF1 in mammals. In response, since ubiquitination at K27 has not been reported, we have removed the data using the triple ubiquitination at K18, K23, and K27.

We have added the following sentence.

Line 316

'These candidate ubiquitination patterns were designed based on known H3 ubiquitination sites mediated by UHRF1 in mammals⁵⁰.'